# Deformation constraints of graphene oxide nanochannels under reverse osmosis

Kecheng Guan [1], Yanan Guo[2], Zhan Li[1], Yuandong Jia[1,3], Qin Shen[1,3], Keizo Nakagawa[1,4], Tomohisa Yoshioka[1,4], Gongping Liu [2], Wanqin Jin [2] & Hideto Matsuyama [1,3]

Nanochannels in laminated graphene oxide nanosheets featuring confined mass transport have attracted interest in multiple research fields. The use of nanochannels for reverse osmosis is a prospect for developing next-generation synthetic water-treatment membranes. The robustness of nanochannels under high-pressure conditions is vital for effectively separating water and ions with sub-nanometer precision. Although several strategies have been developed to address this issue, the inconsistent response of nanochannels to external conditions used in membrane processes has rarely been investigated. In this study, we develop a robust interlayer channel by balancing the associated chemistry and confinement stability to exclude salt solutes. We build a series of membrane nanochannels with similar physical dimensions but different channel functionalities and reveal their divergent deformation behaviors under different conditions. The deformation constraint effectively endows the nanochannel with rapid deformation recovery and excellent ion exclusion performance under variable pressure conditions. This study can help understand the deformation behavior of two-dimensional nanochannels in pressure-driven membrane processes and develop strategies for the corresponding deformation constraints regarding the pore wall and interior.

Several discoveries have been reported concerning nanoscale confined geometric space[1]. Nanoconfinement alters the thermodynamics and kinetics of the species inside, affecting anomalous phenomena and remarkable performance in various applications, such as chemical sensing[2], catalysis[3], energy storage[4], and separation[5]. Owing to the development of nanotechnology, such confinement can be achieved with different dimensions in various materials, e.g., metal-organic frameworks, carbon nanotubes, and nanosheets provide zero-, one-, and two-dimensional cavities or channels at the sub-nanometer and nanometer scales. Fluid transport confined in nanometric dimensions has drawn significant attention[6–11] as it displays distinct behaviors that are considerably different from those in a bulk system. Multiple

factors, including the chemical nature and dimensionality of the confining nanopore interacting with the altered properties of the confined fluid, may lead to unexpected liquid transport[12]. This is particularly relevant for improved separation at the molecular scale, exhibiting significant potential for rapid transport and precise differentiation.

Nanochannel-confined transport can be easily performed in the interlayers of laminates using stacked two-dimensional (2D) materials, such as graphene oxide (GO) nanosheets. Multilayered GO nanosheets form a heterogenous network of angstrom-scale voids between the sheets, which function as membrane nanochannels for transport and separation. This nanochannel separation membrane can be fabricated by a practically feasible way as compared to other techniques, such as

[1]Research Center for Membrane and Film Technology, Kobe University, 1-1 Rokkodai, Nada, Kobe 657-8501, Japan. [2]State Key Laboratory of Materials-Oriented Chemical Engineering, College of Chemical Engineering, Nanjing Tech University, 30 Puzhu Road (S), Nanjing 211816, China. [3]Department of Chemical Science and Engineering, Kobe University, 1-1 Rokkodai, Nada, Kobe 657-8501, Japan. [4]Graduate School of Science, Technology and Innovation, Kobe University, 1-1 Rokkodai, Nada, Kobe 657-8501, Japan. ✉e-mail: wqjin@njtech.edu.cn; matuyama@kobe-u.ac.jp

nanopore drilling[13]. Additionally, the feasibility of local structure control[14] imparts GO membrane nanochannels the advantage of tunability in both physical dimensions and chemical environments. However, the consistency of nanochannel confinement in response to different external conditions remains questionable because the hydration of GO easily deforms its bulk form[15] when an external force or pressure is applied. Moreover, owing to the translocation of stacked nanosheets in hydrated laminates, nanochannel confinement is vulnerable in aqueous environments[16]. Correspondingly, most of the reported GO nanochannel membranes are used for nanofiltration to exclude divalent ions and fail to prevent the permeation of smaller species, including monovalent ions[17]. The charge effect from the functional groups of GO nanosheets[18] is effective for the exclusion of species with high charges, whereas for eliminating weakly charged or neutral species, nanoconfinement plays a vital role[19].

The confinement of nanochannels to minimize deformation[20–23] and maximize interactions with target species[24,25] has been investigated. The nanochannel sizes obtained using several characterization techniques have been claimed to be constant to demonstrate the separation mechanism of size exclusion. However, the characterization is not typically performed in situ, and, therefore, is not representative of the actual separation process. Furthermore, the channel size tends to either expand or collapse as external conditions change, leading to ambiguous situations differing from the characterized results, which is a major concern[26,27]. Nanochannels with the same channel size may also exhibit disparate stability when external conditions change owing to different channel functionalities. Hence, investigating the confinement effects of the nanochannels is necessary. Additionally, while pressure-driven filtration is widely used for separation evaluation of GO membrane nanochannels, only the equilibrium states of the nanochannels are typically reported, and their dynamic characteristics have received less attention. Typically, pressurized processes, such as reverse osmosis (see illustration in Fig. 1), require high pressures, necessitating the robust tolerance[28] of the nanochannels formed in the laminates (Fig. 1) to minimize perturbed structure-rendered low salt rejection. Hence, it is vital to understand the response and dynamic behavior of nanochannels while transporting pressure-driven flows.

To investigate the role of inconsistent nanoconfinement in nanochannel membranes during typical pressure-driven desalination, we prepared GO-membrane nanochannels with different confinement susceptibilities and demonstrated their feasibility for ion exclusion. A small, hydrophilic reducing sugar molecule, glucose (Glc), was selected as an agent for the chemical conversion of GO[29] because it is effective in controlling nanosheet chemistry and is likely to be confined in the interlayers of stacked nanosheets[30] without significantly impacting the channel dimensions during assembly. Therefore, we investigated the effects of the pore wall and interior chemistry of the nanochannel on nanoconfinement as well as the influence of nanoconfinement on water and ion transport.

## Results

### Controlling GO membrane nanochannels

The chemical reduction of GO consumes its oxygenated functional groups and generally contaminates the nanosheet with partial reactants. The stacking behavior of the reacted GO nanosheets depends on both the reduction of the GO and intercalation of the reactant molecules. Reducing agents exhibiting favorable interactions with GO are preferred to regularize GO stacking and form nanochannels. In this study, Glc, which does not exhibit solid interplay with GO, was used as the reducing agent to observe the trade-off between GO reduction and Glc intercalation. GO reduction removes its oxygenated functional groups and narrows the dimensions (improves the confinement) of the interlayered nanochannel between neighboring nanosheets, whereas the possibly intercalated Glc in the laminates induces an opposite effect on the channel dimensions (Fig. 2a). Chemical reactions between GO and Glc (producing Glc-rGO) were performed in the presence of ammonia[31] in an aqueous environment (Supplementary Figs. 1 and 2) to remove the oxygenated functional groups of GO (Fig. 2b shows the reduction of the hydroxyl group in GO as an example). Laminates comprising nanochannels were then formed by the filtration-directed assembly[32] of nanosheets (Supplementary Figs. 3–5). With a prolonged reaction period and the increased weight ratio of Glc to GO, the oxygen components in the Glc-rGO were continuously removed, as observed in the X-ray photoelectron spectroscopy (XPS) results of the stacked laminates (Figs. 2c and 2d and Supplementary Figs. 6 and 7). The O/C ratio of Glc-rGO decreased with an increase in the Glc weight ratio, implying that the intercalated content of Glc in laminates was limited to increase the overall O/C ratio.

To investigate the Glc intercalation, X-ray diffraction (XRD) analysis was performed on pristine GO and Glc-rGO laminates produced under different reaction conditions, i.e., with varying reaction periods (T0–T80, representing a reaction period of 0–80 min), and Glc ratios (Glc0–Glc20, representing the weight ratio of Glc to GO of 0–20). The laminate sample is termed Glc'x'T'y' where 'x' and 'y' denote the ratio of Glc to GO and reaction period, respectively. The position shift of the GO characteristic peaks (d001 reflection) indicated in the XRD spectra was inconsistent with the change in reaction conditions (Fig. 2e and Supplementary Table 1). With an increase in the reaction period, the peak shifted to higher $2\theta$ values until T40 and then to lower $2\theta$ values for T60 (the peak intensity of T80 is too low to assign). With an increasing Glc ratio, the peak position did not significantly change from Glc1 to Glc15 (the peak intensity of Glc20 was too low to assign). Clearer characteristic peaks could be observed in laminate samples with increased deposition amount, and similar peak shifting

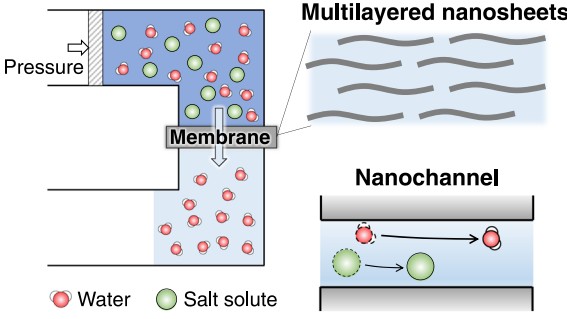

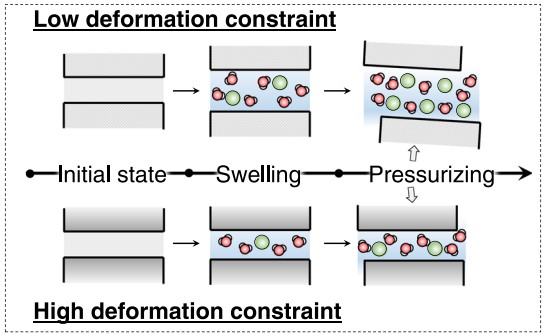

**Fig. 1 | Illustrations of reverse osmosis system, multilayered nanosheet membrane, and interlayered nanochannels.** During reverse osmosis, external pressure is applied to force water molecules through a semipermeable membrane. Nanochannels are formed by neighboring nanosheets in the multilayered nanosheet membrane. Different perturbation of nanochannels with different deformation constraint occurs in the pressurized flow of water and salt.

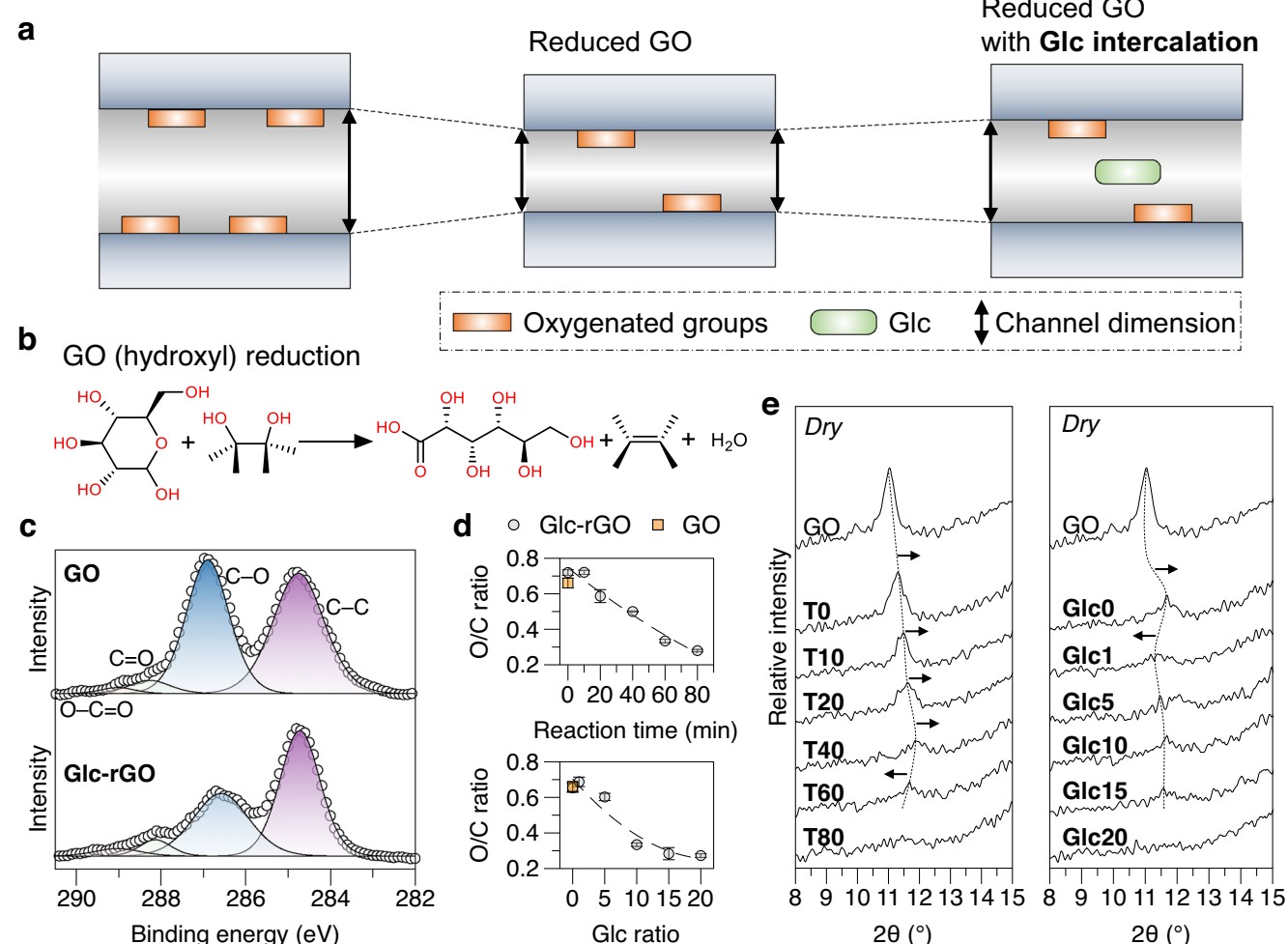

**Fig. 2 | Fabrication of different GO membrane nanochannels. a** Illustration of varied GO membrane nanochannel dimensions by reaction with Glc; **b** Chemical reaction of GO (hydroxyl) reduction by Glc; **c** XPS spectra of GO and a typical Glc-rGO sample (reaction period, 60 min; Glc ratio, 10); **d** O/C ratio of GO and different Glc-rGO samples from XPS analysis. The error bars represent the standard deviations of three measurements; **e** X-ray diffraction spectra of GO and Glc-rGO samples (dashed lines and arrows indicate peak shifts). The Glc ratio for the Glc-rGO samples derived from different reaction periods (T0–T80) was fixed at 10 (Glc10), whereas the reaction period for the Glc-rGO samples derived from different Glc ratios (Glc0–Glc20) was fixed at 60 min (T60). Source data are provided as a Source Data file.

tendencies with a change in the reaction conditions were confirmed (Supplementary Fig. 8). These results tentatively reveal the Glc intercalation effect during nanosheet stacking, as a higher reduction degree is typically accompanied by a continuous peak shift to higher 2θ values[19]. Although the same amount of Glc molecules was used for T'*y*' membrane, different reaction periods induced different reduction degree of GO, which affected the interactions between the Glc and the nanosheets and nanosheet stacking order. A higher reduction degree of GO may induce worse compatibility with the Glc molecules, causing unfavorable stacking during filtration-induced assembly and increasing the overall interlayer spacing. A higher amount of Glc used would lead to a higher reduction of GO in the Glc'*x*' membranes, decreasing the interlayer spacing. However, higher amount of Glc may also have a higher possibility of retaining more Glc molecules, increasing the interlayer spacing. Consequently, this trade-off may restrict the interlayer spacing change, maintaining a similar peak position, as shown in the XRD spectra. The shift in the peak position was limited to the narrow range of 11–12° (Supplementary Table 1), implying that the dimension change of the nanochannel after the reaction was insignificant. Therefore, investigating the impact of minor changes in the nanochannel on channel properties under reverse osmosis conditions would be beneficial.

## Nanochannel confinement in water

The interlayer conditions may differ when the laminates are immersed in water. The hydrophilicity of the GO nanosheets allows water molecules to adsorb onto the basal plane, which leads to water intercalation between neighboring nanosheets in the laminates[33]. This process is similar to polymer swelling, which occurs with favorable interactions between polymer chains and solvent molecules[34]. The resulting expansion of the interlayered nanochannels may prevent effective ion sieving. Additionally, when the wet nanochannel was subjected to a pressurized water flow, the channel could be perturbed with increased permeance but decreased salt rejection. We immersed the substrate-supported GO and Glc-rGO laminates into water to preliminarily check their water stability (Supplementary Fig. 9). The Glc-rGO laminates remained integrated while the GO laminates were seriously destructed by the water intrusion, confirming the improved macroscale stability of Glc-rGO. Therefore, through the chemical conversion process by Glc, GO laminates could be more sufficiently stable to have confined nanochannels for ion separation. As Glc could reduce GO and was likely to be inserted in the laminates, hence, the roles of Glc in chemically converted GO laminates were further studied.

Molecular dynamics (MD) simulations were performed to investigate the evolution of the laminate structure under water pressure

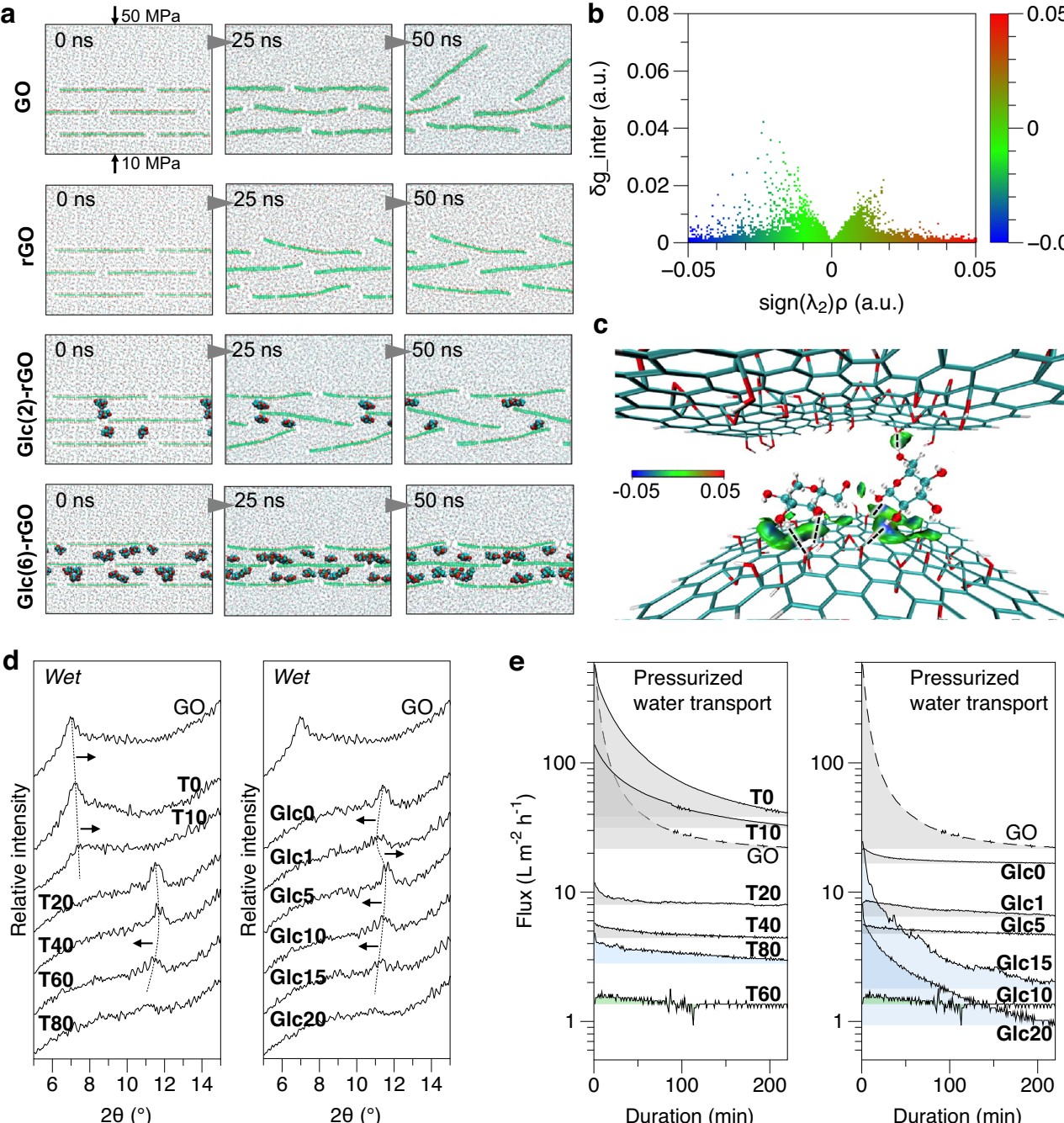

**Fig. 3 | Nanochannel deformation in water. a** MD simulations of laminate models of GO, rGO, and Glc-rGO in pressurized water. Pressures of 50 and 10 MPa were applied on the opposite sides of the simulation cell for all the models, as indicated in the GO model as an example. Two Glc-rGO models with two and six Glc molecules in each channel were constructed, denoted as Glc(2)-rGO and Glc(6)-rGO, respectively; **b** sign($\lambda_2$)$\rho$ mapped 2D scatter plot of the intermolecular electron density gradient difference ($\delta$g_inter). The value of sign($\lambda_2$)$\rho$ is represented by filling color according to the color bar. $\rho$ denotes the electron density at the weak interaction critical point, sign($\lambda_2$) denotes the sign of the second largest eigenvalue

$\lambda_2$ of the electron density Hessian matrix; **c** Isosurface of $\delta$g_inter with an isovalue of 0.005 a.u. The blue isosurface between the oxygen-containing group of rGO and the hydroxyl group of Glc indicates the C-H...O hydrogen bond interactions as depicted by dashed lines; **d** XRD spectra of wetted GO and Glc-rGO samples (dashed lines and arrows indicate peak shifts); **e** Time-course pure water flux of GO and Glc-rGO samples at 1 MPa. Glc ratio for Glc-rGO samples derived from different reaction periods (T0–T80) was fixed at 10 (Glc10), whereas the reaction period for Glc-rGO samples derived from different Glc ratios (Glc0–Glc20) was fixed at 60 min (T60). Source data are provided as a Source Data file.

conditions and to reveal the contribution of Glc to the structural stability (Fig. 3a, Supplementary Fig. 10, and Supplementary Movies 1–4). rGO laminates with lower O/C ratios exhibited a more stable structure during a simulation of 50 ns, compared to GO, which is ascribed to the increased hydrophobicity of rGO. This evidences the positive effect of chemical reduction. It was also found that the inserted Glc in the interlayers had a positive effect on improving the laminate stability, as

the same rGO laminate model with more of intercalated Glc molecules exhibited better stability from the simulation results (Fig. 3a, Glc(2)-rGO and Glc(6)-rGO). In this regard, the inserted Glc should have certain interaction with GO laminates. To identify the nature of the interaction between the Glc and rGO that resulted in the favorable intercalation, a cluster model (Supplementary Fig. 11) from the last frame of the MD simulation trajectory was employed by applying the

independent gradient model (IGM) method. The 2D scatter plot and the isosurface (sign($\lambda_2$)$\rho$ mapped intermolecular electron density gradient difference ($\delta g\_inter$)) for the model are shown in Fig. 3b. The region close to sign($\lambda_2$)$\rho = 0$ (green) denotes mainly the weak attractive van der Waals interactions. The cyan-to-blue region denotes relatively strong attractive interactions, including hydrogen bonds (Fig. 3c). It indicates that the hydrogen bonding was the main interaction between Glc and GO for their integration in the laminates. In addition, the chemical reduction role and hydrogen bonding role of Glc were also experimentally evidenced by observed absorption peak shift in ultraviolet–visible (UV-vis) spectra and chemical shift in XPS spectra of GO and Glc mixture (Supplementary Fig. 12). As a result, Glc molecules could be favorably retained in laminates and adsorbed by laminates as observed in corresponding experiments (Supplementary Fig. 13).

In practical chemical conversion process, using same amount of Glc but increasing reaction period (T'$y$' membranes) will produce higher reduction degree of GO, which limits the hydrogen bonding sites. Similarly, using same reaction period but increasing used amount of Glc (Glc'$x$' membranes) also has similar issue. In these cases, Glc might have adverse swelling effects on the nanochannels because of its hydrophilicity. In the reduction of GO via Glc, the obtained rGO should have an improved hydrophobicity for the pore wall of the nanochannel, whereas the included Glc imparts hydrophilicity to the pore interior. This balance is important for constraining the deformation of the nanochannel under aqueous and pressurized conditions. Under wet conditions, notable differences in the characteristic peak positions in the XRD spectra were observed for different laminates (Fig. 3d and Supplementary Table 1). The GO, T0, and T10 laminates shifted to lower 2θ values of ~7° from the original position of 11–12° (Fig. 2e), indicating significant water intercalation into the interlayered channels. Better water-wetting properties (lower water contact angles) were obtained for the GO, T0, and T10 laminates than for the other samples (Supplementary Fig. 14). Hence, water penetrates easier in the interlayers of these laminates, increasing the interlayer spacing. For other samples exhibiting higher water contact angles, a similar XRD peak position was maintained as that under dry conditions, but the peak shifted with an increased reaction period (T60 or higher) and Glc ratio (Glc10 or higher), evidencing the adverse aspect of Glc. Because the nanochannel is framed by the hydrophobic rGO nanosheets, it can prevent water intercalation to a certain extent when no external driving force is applied. Therefore, the peak shift (nanochannel instability) caused by Glc in the nanochannels may have been underestimated in the XRD characterization of the simply wetted samples.

To further investigate the nanochannel deformation in laminates with the infiltrated pressurized flow (which is similar to pressure-driven filtration), the time-course pure water flux of different laminates at an applied pressure of 1 MPa was recorded (Fig. 3e). With an externally driven force, water molecules can easily penetrate the nanochannels, resulting in a higher propensity for channel deformation. Rapid increases and subsequent decreases in the flux were observed for the GO, T0, and T10 samples, correlating with the XRD analysis (Fig. 3d). With longer reaction periods and higher Glc ratios, the flux decline was gradually minimized up to a reaction period of 60 min (T60) or Glc ratios of 10 (Glc10). For T80, Glc15, and Glc20, the decline again became obvious. This phenomenon is in accordance with our hypothesis that the included Glc molecules could be another cause of nanochannel deformation during pressurized filtration. Higher Glc ratio resulted in a more significant flux decline (samples Glc15 and Glc20 with higher Glc ratios used for laminate preparation in comparison to T80 with longer reaction periods) because the number of intercalated Glc molecules in the nanochannels would increase for Glc15 and Glc20. The optimal samples (Glc10 or T60) exhibited a minimal water flux decline, owing to the nanochannel constraint caused by the optimal integration of GO frameworks and Glc

intercalations. Both the occupation of Glc molecules in the channels and the reduced interlayer spacing hindered water transport (simulation results in Supplementary Fig. 15 and Supplementary Movies 5 and 6), resulting in a decreased water flux, particularly when including a possibility of more Glc molecules in the nanochannel.

Although the observed swelling and compaction phenomena of the GO nanochannel membrane are similar to those of the polymeric membranes, the structural changes after swelling are significantly different. 2D material laminates have well-defined nanochannels, which are quite different from the free volume generated by the chain mobility in polymers. The factors affecting the structural stability of laminates could thus be distinct, as we observed that the chemistry of both the nanosheet and confined molecules between nanosheets had an impact. Determining the properties of laminates is important for constructing stable and tunable nanochannels for studying the mass transport and performance improvement in nanofluidic applications.

## Nanochannel confinement in salt/water mixture

The cations and their hydrated forms significantly affect the structural stability of the GO laminates. In particular, the nanochannel deformation of laminates in aqueous environments is significant in the presence of sodium ions (Na$^+$). The large hydration radius of Na$^+$ expands the nanochannel dimension, and its weak attraction with GO cannot compensate for this deformation, leading to structural instability in aqueous solutions containing Na$^+$[35]. Consequently, the nanochannel deformation behavior in the NaCl solution was further investigated.

Studies using pure water-wetted laminate samples for stability characterization may underestimate the nanochannel deformation when applied in desalination processes to retain sodium salts, including Na$_2$SO$_4$ and NaCl. For a comparison of the dry and pure water-wetted samples characterized using XRD (Figs. 2e and 3d, respectively), samples wetted with NaCl solution (500 ppm) were also analyzed (Fig. 4a and Supplementary Table 1). In contrast to the virtually unchanged peak positions under water-wetting conditions for the T20, T40, Glc0, Glc1, and Glc5 samples, they exhibited a notable shift to a lower 2θ value, indicating a greater deformation effect of the NaCl solution than that of pure water on the samples.

We hypothesized that samples constraining the nanochannel deformation to varying degrees would result in varying water/salt separation performances. The Glc10T60 sample was expected to provide optimal results from the above XRD analysis and time-course flux evaluation. Pre-compaction of these membranes by pure water filtration was applied before further performance evaluations, particularly for the pristine GO membrane (Supplementary Fig. 16). The pressure-driven filtration results with Na$_2$SO$_4$ and NaCl solutions for different samples are depicted in Fig. 4b, exhibiting the optimal point (Glc10T60) for the rejection performance of both salts, agreeing with the time-course flux evaluation results. The NaCl rejection was optimized to >90% using the optimal membrane (Glc10T60) with a favorable nanochannel deformation constraint.

To further investigate laminar membranes with different nanochannel deformation constraint capabilities, four typical samples were selected for comparison: pristine GO, Glc10T20 (optimal Glc ratio but short reaction period), Glc10T60 (optimal), and Glc20T60 (optimal reaction period but excessive Glc ratio). The surface zeta potentials of these membranes at neutral pH were all negative (Fig. 4c and Supplementary Fig. 17), which is desirable for the rejection of salt solutes containing high-valence anions (such as Na$_2$SO$_4$). The rejection of different salts by these membranes (Fig. 4d) indicates that the ionic transport behavior is governed by the charge effect, leading to a high rejection of Na$_2$SO$_4$ and low rejection of MgCl$_2$. However, the difference in rejection by Glc10T60 and Glc20T60 between the different salts narrowed significantly compared with that of GO and Glc10T20. The lower negative zeta potentials of Glc10T60 and Glc20T60 could

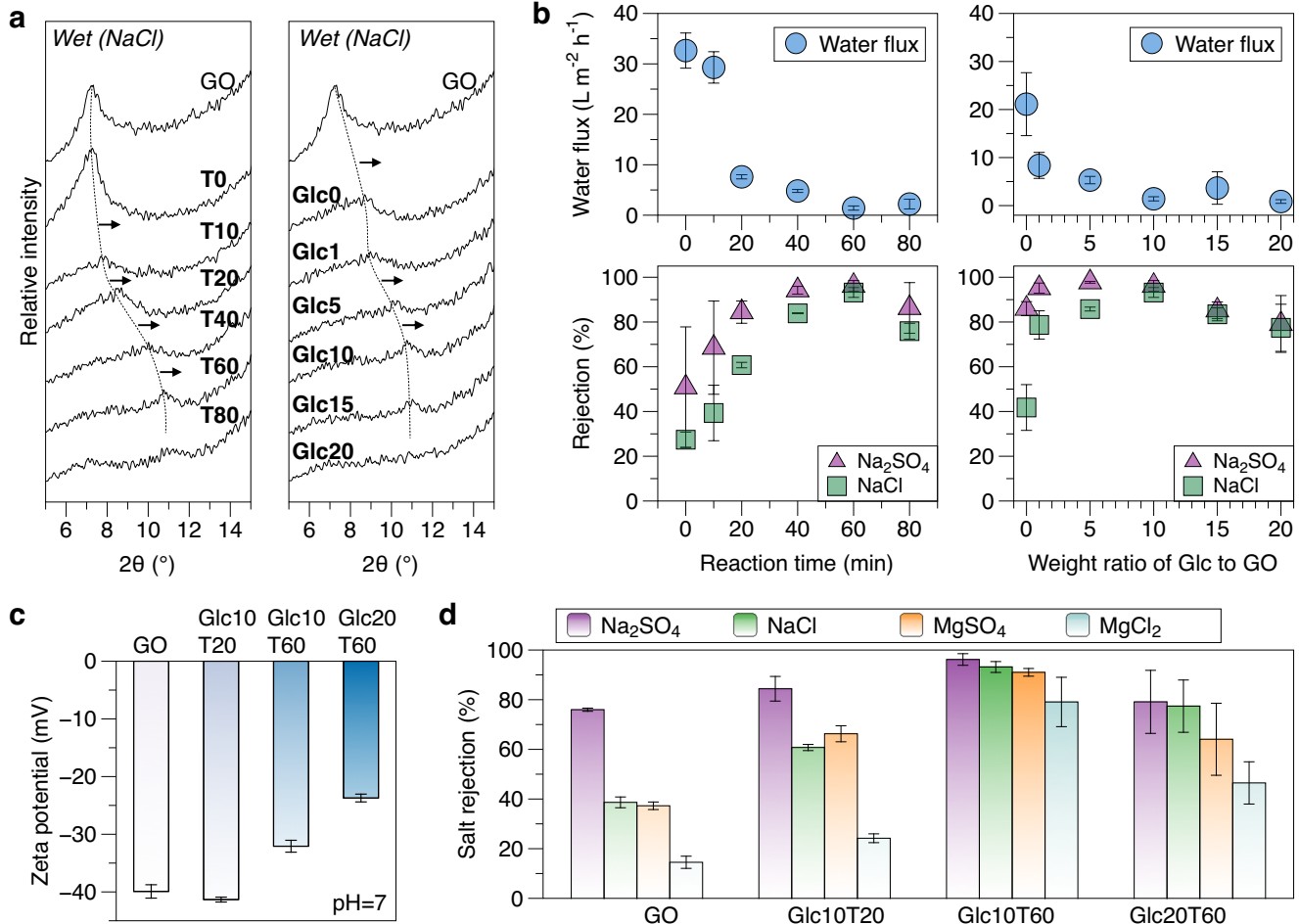

**Fig. 4 | Nanochannel deformation in salt solution and reverse osmosis performance. a** XRD spectra of GO and Glc-rGO samples wetted by 500 ppm NaCl solution (dashed lines and arrows indicate peak shifts); **b** Separation performance of Glc-rGO membranes; **c** Surface zeta potentials of GO, Glc10T20, Glc10T60, and Glc20T60 membranes at pH = 7; **d** Rejection of salts by GO, Glc10T20, Glc10T60, and Glc20T60 membranes. The error bars represent the standard deviations of three measurements for rejection performance and surface zeta potential. Source data are provided as a Source Data file.

explain their decreased attraction to high-valence cations, improving MgCl₂ rejection. Notably, stronger nanochannel confinement (an optimal example is Glc10T60) should improve the overall rejection of all types of salts by steric hindrance. However, the excessive use of Glc molecules (Glc20T60) resulted in unfavorable nanochannels for the separation and compromised salt rejection, indicating the adverse effects of the excessive amount of Glc used on the membrane performance. Our previously proposed nanochannel-confined charged repulsion[19] for the GO nanochannel membrane emphasized the role of nanochannel confinement. Additionally, in this study, we determined the significance of the confinement constraint capability. Careful consideration should be given to the possible disruption of channeled nanosheets by intercalated substances.

### Role of the deformation constraint of nanochannel for salt/water separation

The deformation constraints of the different GO membrane nanochannels were analyzed through continuous filtration cycles of the NaCl solution with pressure variations (1 MPa → 2 MPa → 1 MPa → 2 MPa → 1 MPa, Fig. 5a–d). After pre-compaction by pure water filtration, when initially fed with NaCl solution at 1 MPa, the GO and Glc10T20 membranes (Fig. 5a and b) exhibited a gradually increasing permeation flux owing to unstable channel confinement. When the applied pressure was increased to 2 MPa, a rapid increase and subsequent decline in the permeation flux were observed for both

membranes (90% and 50% for GO and Glc10T20, respectively), with a decreased and gradually recovered NaCl rejection. The permeation flux significantly stabilized after pressure variation, indicating the deformability of membrane nanochannels in GO and Glc10T20 membranes under external pressure conditions. With the improved deformation constraint, the Glc10T20 membrane exhibited a higher and more stable NaCl rejection than GO during the filtration process.

The Glc10T60 membrane with optimal nanochannel stability achieved the highest and most stable NaCl rejection (~90%) during filtration with pressure variation, although its flux decline at increased pressures was higher than that of Glc10T20 (Fig. 5c). However, the Glc20T60 membrane performed poorly with a NaCl rejection similar to that of Glc10T20 (~60%) but with a significantly lower permeation flux (Fig. 5d), although these laminates remained intact after pressurized filtration (Supplementary Fig. 18). The filtration performances of different membranes at the 1 MPa steps in the filtration cycle (three 1 MPa steps in the testing cycle of 1 MPa → 2 MPa → 1 MPa → 2 MPa → 1 MPa) were compared (Supplementary Fig. 19) to investigate irreversible changes in the membrane properties. If the membrane structure is sufficiently stable, the observed flux and salt rejection performance should be similar under the same external conditions. However, NaCl rejections for the GO, Glc10T20, and Glc20T60 samples were compromised after the pressurization process, implying that certain structural deformations were applied during the pressurization and depressurization. The deformation could be the displacement of

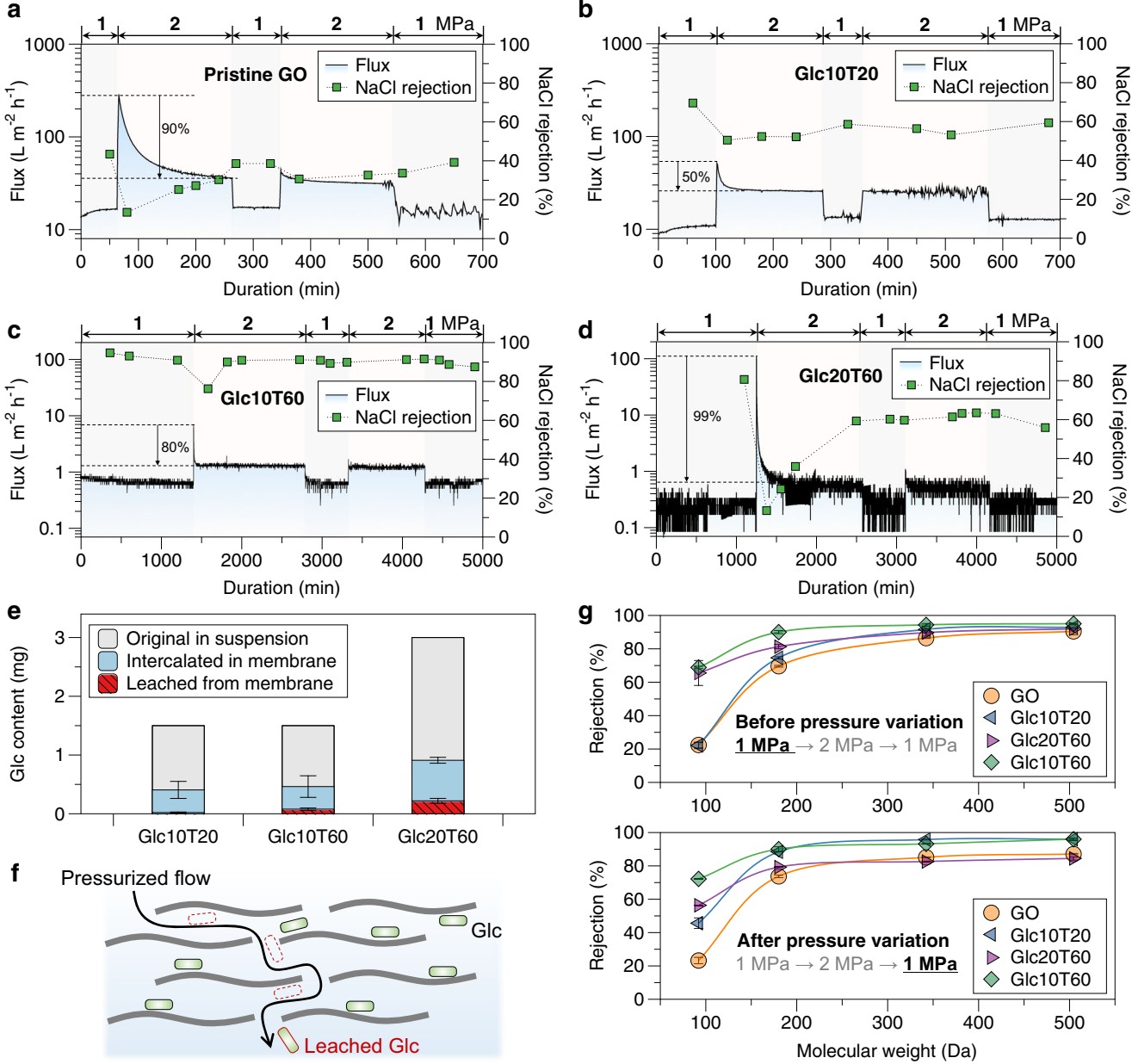

**Fig. 5 | Nanochannel deformation and performance in pressure-changing reverse osmosis.** Filtration of NaCl solutions with pressure variations by (**a**) GO, (**b**) Glc10T20, (**c**) Glc10T60, and (**d**) Glc20T60; (**e**) Glc content in original membrane precursor suspension, intercalated Glc content in prepared membranes, and leached Glc content from membranes during permeation test. The error bars

represent the standard deviations of three measurements; (**f**) Illustration of Glc leaching from laminates with the pressurized flow; (**g**) Molecular weight cut-off curves of membranes before and after pressure variations. The error bars represent the standard deviations of three measurements. Source data are provided as a Source Data file.

nanosheets in their laminates by swelling and compaction processes, altering the nanochannel confinement.

The improved hydrophobicity of these three Glc-rGO membranes (Supplementary Fig. 14) indicates the improved water stability of their bulk structures. This was inconsistent with their nanochannel stability and salt rejection, as they had different responses to the infiltrated water flow in their different pore interiors. The amount of Glc retained after the preparation of these laminates could be different, considering the different degrees of GO reduction and interactions with Glc. The amount of Glc intercalated was then quantified and result was similar for the Glc10T20 and Glc10T60 laminates, whereas it was higher for Glc20T60 (Fig. 5e and Supplementary Fig. 13a), indicating that higher amounts of Glc used in suspension lead to a higher possibility of retained Glc in the corresponding laminates. The higher Glc content in Glc20T60 laminates may be associated with its relatively

poor performance. Although this explains the difference in performance between Glc10 and Glc20, the difference between T20 and T60 must be further investigated.

As Glc molecules have hydrophilic properties, the intercalated Glc molecules can also be dissolved with pressurized flow during the permeation test (Fig. 5f). The leached amounts of Glc for these membranes during permeation tests were quantified (Supplementary Fig. 20) in the order of Glc10T20 < Glc10T60 < Glc20T60 (5%, 15%, and 24%, respectively; Fig. 5e). This order is also consistent with their flux decline tendency shown in Fig. 5b–d, indicating that the presence of Glc molecules in the nanochannels was related to channel stability. Higher Glc leaching indicates lower integration of Glc and rGO and, thus, a higher possibility of a perturbed structure during Glc leaching by pressurized flow. Therefore, the intercalation amount of Glc and the integration of Glc and GO for the laminates are influential for the

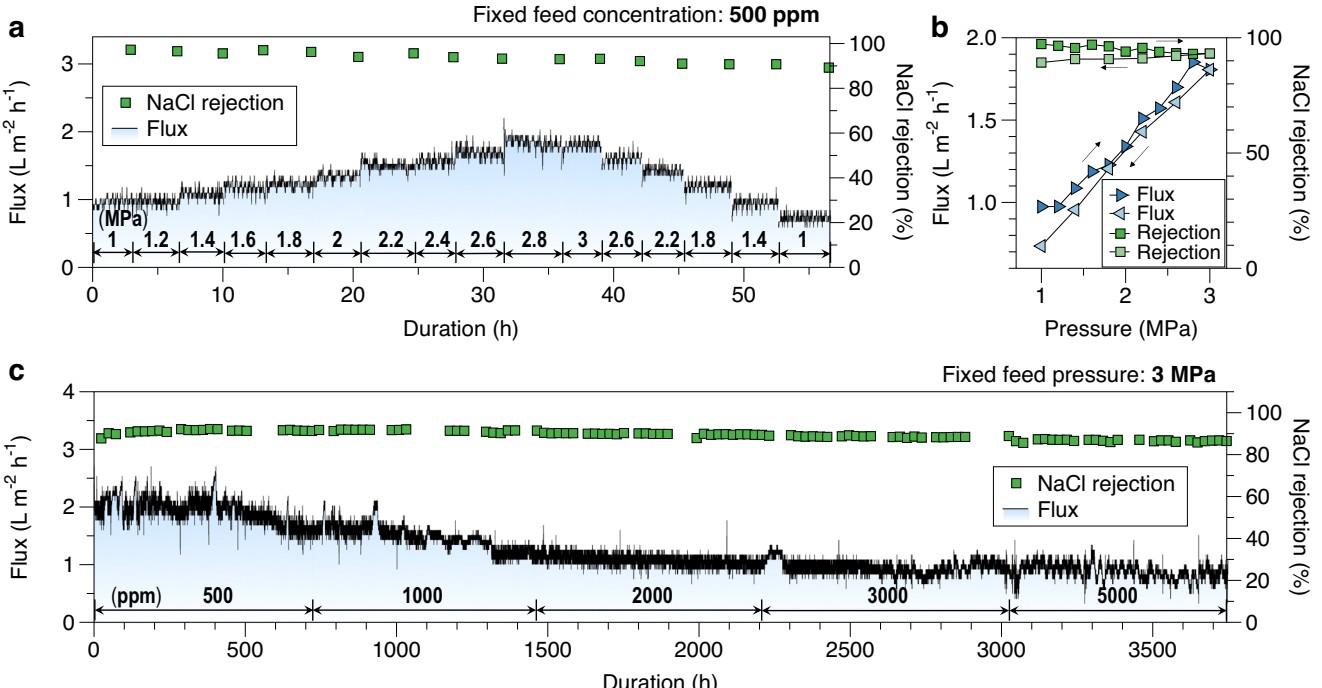

**Fig. 6 | Filtration performance durability of the Glc10T60 membrane.**
**a** Filtration of 500 ppm NaCl solution with stepwise changes in the operating pressures; **b** Changes in flux and NaCl rejection during increasing and decreasing pressures (arrows indicate the pressure increase and decrease); **c** Long-term filtration of NaCl solution with changed concentration (500 ppm → 5000 ppm) at 3 MPa. Source data are provided as a Source Data file.

nanochannel stability and membrane performance. The mutable and dynamic GO nanochannels cannot function like rigid pores in response to external pressure and NaCl in the feed, to strictly exclude hydrated ions from water. Nanochannel deformation frequently occurs when external conditions are changed. An improved deformation constraint can be obtained by optimizing the chemistry of the GO nanosheets and avoiding the excessive intercalation of undesirable molecules into multilayered nanochannels to maximize the stable rejection of ions.

The molecular weight cut-off (MWCO, defined as the lowest molecular weight solute in which the membrane retains 90% of the solute) of these membranes was investigated to reveal their separation capability toward neutral solutes and channel dimensions before and after the pressure variations. Rejection tests for organic solutes with a suitable molecular weight[36], including glycerol (92 Da), Glc (180 Da), sucrose (342 Da), and raffinose (504 Da), were conducted to obtain MWCO curves (Fig. 5g). Before the pressure variation, the MWCO values of the different membranes were ~180–500 Da, and the order corresponded to the NaCl rejection results. After pressure variations, nanochannel deformation occurred as the MWCO values of GO, Glc10T20, and Glc20T60 changed, whereas the optimal Glc10T60 remained stable (Supplementary Table 2). GO and Glc20T60 were transformed into less confined nanochannels, whereas Glc10T20 improved. The pore radius distributions of GO and Glc10T20 (Supplementary Fig. 21) were attainable according to their MWCO curves, where the GO exhibited similar results after the pressurization, whereas Glc10T20 exhibited a narrower distribution and smaller pore radius. This indicates that pressurization can either improve or deteriorate the deformable nanochannel properties for separating molecules within a certain size range. From the results of flux decline at high pressures (Fig. 5a–d), the Glc10T20 membrane exhibited the optimal performance, as it achieved moderate deformation constraints by limited intercalation and leaching of Glc molecules inside the nanochannels (Fig. 5e). Hence, it was confirmed that pressurization can improve

the packing of multilayered structures, and in some cases, improve GO nanochannel membranes to render higher separation performance[22]. In contrast, the nanochannels of GO- and Glc20T60-contained nanosheets with a wider deformation range were sensitive to external conditions and likely to be disrupted in a disordered manner and form defects. The XRD spectra of these laminates before and after the pressurized filtration test thus were compared (Supplementary Fig. 22). A significant decrease in the peak intensity was observed for pristine GO, whereas a similar intensity but smaller interlayer spacing (peak shifting to higher 2θ values) was observed for Glc10T20 and Glc10T60 due to pressurization effect or leaching of inserted Glc. The results and rejection performance toward NaCl and organic solutes further confirmed divergent structural changes in mutable GO-based laminates. To fabricate a stable nanochannel membrane with high performance in reverse osmosis, the consideration and evaluation of the deformation constraint is vital, as minor changes in the nanochannel dimensions can result in major differences in reverse osmosis performance.

The Glc10T60 membrane with the optimal salt rejection performance was subjected to additional filtration tests to evaluate its performance. Stepwise increases and decreases in pressure were applied when filtrating a 500 ppm NaCl solution (Fig. 6a). When the pressure increased from 1 to 3 MPa in steps of 0.2 MPa and decreased from 3 to 1 MPa in steps of 0.4 MPa, the membrane maintained a relatively stable NaCl rejection of >90%. However, the membrane fluxes and NaCl rejection at the same pressure during the pressure increase and decrease were different (Fig. 6b), indicating that the nanochannel had deformed, but only to a minor degree. Long-term performance evaluation (>150 days) at 3 MPa was conducted using various NaCl feed concentrations ranging from 500 to 5000 ppm (Fig. 6c). Each feed concentration was tested for ~1 month, and the membrane exhibited stable NaCl rejection throughout the entire period. The membrane flux gradually decreased because of the increased osmotic pressure at high NaCl concentrations and possible membrane compaction. However, the

optimal membrane, possessing a strong constraint on nanochannel deformation, demonstrated promising results. While this study aims to investigate the nanochannel deformation using Glc possessing no distinct interactions with the GO, the obtained stable NaCl rejection performance is still comparable to studies on GO-based membranes (Supplementary Table 3). The findings of this study could aid in designing modifications and intercalations for GO membranes to achieve nanochannel deformation constraints and satisfactory flux and salt rejection. For example, grafting agents that possess distinct chemical structures and functional groups (such as π-conjugated molecules with water-affinity functional groups) can stabilize the nanochannel and favor water diffusion, which should work better than the weakly interacted Glc demonstrated in this study.

## Discussion

In summary, we investigated the nanochannel deformation behavior of multilayered GO membranes and determined the balance between the channel chemistry and confinement stability for the reverse osmosis of salt solutions. The deformation depended on the pore wall (GO nanosheets) and pore interior (intercalated moieties) of the nanochannel. Notably, while minor changes to the pore interior negligibly altered the channel dimensions from typically characterized values, they may result in a significant difference in the nanochannel stability (deformation constraint) during reverse osmosis. Thus, the discord between the characterization and performance of the GO nanochannel membranes was highlighted. The effect of the intercalation chemistry should be carefully considered when designing nanochannel membranes using functionalized nanosheets to construct confined nanochannels. There is still a considerable performance gap between the state-of-the-art GO-based membranes and commercial polyamide membranes for reverse osmosis. Therefore, future efforts are required to improve the quality of raw GO materials and search for effective modification agents. Suitable modification agents that can constrain the swelling or compaction of laminates and exhibit low transport resistance in nanochannels are desirable for maintaining constrained nanochannels for precise ion separation. In addition, large-scale production of high-quality GO nanosheets is expected in future to make the fabrication of few-layered laminates more practical for maximizing water flux.

The phenomenon observed in the GO membrane nanochannels under reverse osmosis could also provide useful information for maintaining channel confinement for materials of other laminates and applications of other nanofluidic processes. Although the material and application investigated in this study are specific, we believe that the results on what affects channel confinement and how to maintain it are also applicable to other systems. For example, the nanochannel deformation and its constraint could be extended to laminate membranes, including MXene (Ti3C2Tx)[37,38], boron nitride[39,40], and carbon nitride[41], where functionalization is necessary to achieve stable nanochannels during pressurized membrane separation processes in water, organic solvents, or gas. In addition to the membrane process, the factors affecting the channel confinement constraint are also relevant for other nanofluidic processes involving aqueous chemistry within nanometric slit pores[6]. For example, nanochannel confinement enabled controllable ion transport in laminates, exhibiting significant potential for salinity gradient power generation[42] in the water-energy nexus, and fluid confinement in laminates boosted radical yields for efficient chemical reactions[43] in advanced oxidation processes. However, it is recommended that the consistency of channel confinement in nanofluidic applications be evaluated by designing experiments based on the characteristics of specific processes, thereby acquiring incremental insights into the nanochannel deformation behavior of different materials and applications.

## Methods

### Preparation of Glc-rGO

A 10 mL aqueous suspension containing GO (0.15 mg), 25% ammonia solution (40 μL), and the calculated amount of Glc was vigorously mixed. Thereafter, the mixture was sealed in a tube and heated to 95 °C for a predetermined duration to complete the reaction. The prepared suspension was then cooled to 25 °C at atmospheric pressure for further use.

### Membrane fabrication

A vacuum filtration process was employed to laminate the GO or Glc-rGO nanosheets onto porous substrate membranes. An aqueous suspension containing 0.15 mg of the nanosheets was diluted with ultra-pure water to a volume of 300 mL and sonicated before vacuum filtration. After the water in the suspension was completely filtered, the substrate membrane containing the deposited nanosheets was dried in a desiccator for at least 24 h before further use. Nylon and polycarbonate substrates were used to support the laminates for the performance evaluation and characterization, respectively.

### Desalination performance evaluation

The membranes were sealed in a homemade cross-flow module with an effective membrane area of ~7.1 cm$^2$ for the pure water permeation and salt rejection tests. The membranes were compacted by pure water filtration at 1 MPa for 16 h before further rejection performance evaluations. Salt solutions containing 500 ppm $Na_2SO_4$, NaCl, $MgSO_4$, and $MgCl_2$ were prepared. A transmembrane pressure of 1 MPa was applied during the cross-flow filtration tests. Membrane performance, including water flux and salt rejection values, was obtained when the observed values became constant over time. Typically, a duration of 8–16 h (depending on the membrane permeability) was applied for the filtration test of each feed solution. For the NaCl rejection test, feed concentrations of 1000, 2000, 3000, and 5000 ppm and applied pressures ranging from 1 to 3 MPa were used.

The mass of the permeate from the cross-flow filtration was continuously monitored using an electronic balance (GF-1000, A&D Co., Ltd., Tokyo, Japan). The permeation flux of the membrane ($P$, L m$^{-2}$ h$^{-1}$) was calculated using the following equation:

$$P = \frac{V}{At} \tag{1}$$

where $V$ (L) denotes the volume of the collected permeate, $A$ (m$^3$) the effective membrane area, and $t$ (h) the filtration duration.

The salt concentrations in the feed and permeate were measured using a conductivity meter (EC-33B, LAQUAtwin, HORIBA Ltd., Kyoto, Japan). The apparent rejection coefficient ($R$) of salt was calculated using the following equation:

$$R = \left(1 - \frac{C_1}{C_0}\right) \times 100\% \tag{2}$$

where $c_O$ and $c_I$ denote the ion concentrations in the feed and permeate, respectively.

### Measurement of MWCO

Rejection tests using an aqueous feed of glycerol (92 Da), Glc (180 Da), sucrose (342 Da), and raffinose (504 Da) solutions were sequentially conducted for the GO and the different Glc-rGO membranes. All the feed concentrations were 500 ppm. The rejection was calculated using Eq. 2. The MWCO curves of the four solutes were fitted using modified Bézier Curves[44] employing two points from the data and two interpolated control points.

To investigate the nanochannel deformation (changes in MWCO) under pressure variation for one membrane sample, the applied

pressure was varied from 1 to 2 MPa and then back to 1 MPa to obtain the rejection values and curves at 1 MPa before and after pressure variation.

## Measurement of nanochannel dimensions

X-ray diffraction (XRD, Ultima IV, Rigaku, Tokyo, Japan) was performed with a step size of 0.02° and a scanning speed of 5° min$^{-1}$ to determine the interlayered channel dimensions of the GO and Glc-rGO laminates. To measure the wet samples, droplets of water or NaCl solution (500 ppm) were gradually added to the surface of the sample until complete surface coverage was achieved. After 10 min of wetting, the water or NaCl solution on the surface was carefully removed using a lab tissue paper and then subjected to XRD analysis.

## Characterization

Ultraviolet–visible spectra (UV-vis, V-650, Jasco International Co., Ltd., Tokyo, Japan) of the GO and Glc-rGO suspensions were recorded in the 200–500 nm range. The morphologies of the membrane samples were observed using field-emission scanning electron microscopy (FE-SEM, JSF-7500F, JEOL Co., Ltd., Tokyo, Japan). X-ray photoelectron spectroscopy (XPS, JPS-9010 MC, JEOL Co. Ltd., Tokyo, Japan) was performed using an Al Kα radiation source (1486.6 eV) to analyze the membrane surface chemistry. The surface zeta potentials of the membrane samples were measured using an electrokinetic analyzer (Surpass 3, Anton Paar, Graz, Austria). The water contact angles of the different membrane samples were analyzed using a contact angle testing system (Drop Master 300, Kyowa Interface Science Co., Ltd., Tokyo, Japan) containing 40 µL water droplets.

## Computational simulation details

To investigate the stability of the laminate structure in water under applied pressure, four models were constructed including GO, rGO, Glc(2)-rGO (two glucose molecules inserted into each channel), and Glc(6)-rGO (six glucose molecules inserted into each channel) (Supplementary Fig. 10). The GO and rGO sheets were built based on the Lerf-Klinowski model[45], in which the hydroxyl and epoxy groups were randomly distributed on both sides of the carbon basal plane. The O/C ratio and interlayer spacing of GO/rGO were determined by matching the values obtained from practical samples. For the GO sheet, the O/C ratio was ~0.6, and the interlayer spacing was 8.03 Å. For the rGO sheet, the O/C ratio was ~0.3, and the interlayer spacing was 7.62 Å. The membrane was composed of trilaminar GO/rGO sheets with a slit of ~6 Å in the $x$–$y$ plane. The offset between the slits of the GO/rGO sheets was 29.5 Å. Different numbers of Glc molecules intercalated in the constructed laminate models were employed to explore the effect of the Glc. Two rigid graphene sheets were placed at the ends of the cell. The space between the two graphene sheets was filled with water. A vacuum space of ~40 Å along the $z$-direction was applied to prevent interactions between the system and its images. Pressures of 50 and 10 MPa were applied to the two graphene sheets. The size of the orthogonal cell of the four models was $61.5 \times 29.8 \times 150.0$ Å$^3$ ($x$, $y$, $z$) (Supplementary Fig. 23a). In the current study, the purpose of applying the unbalanced force (that is, the nonequilibrium MD simulation) is to mimic the condition in which the membrane is squeezed under the pressure and then investigate the behavior of the membrane during this process. Similar nonequilibrium MD simulations have been successfully employed by applying a higher pressure on one side than on the other to deal with various membrane related topics[46–49].

To investigate the water transport through the laminates of GO and Glc-rGO, two models were constructed: GO and Glc(6)-rGO (six Glc molecules inserted into each channel) (Supplementary Fig. 15). In the two models, an orthogonal system cell of $61.5 \times 29.8 \times 119.1$ Å$^3$ ($x$, $y$, $z$) was used with periodic boundary conditions applied in all directions. The membrane comprised trilaminar GO/rGO sheets with a slit of ~6 Å in the $x$–$y$ plane. The offset between the slits of the GO/rGO sheets was

29.5 Å. One rigid graphene sheet was placed 40 Å from the nearest GO/rGO sheet. The chamber between the graphene sheet and the nearest GO/rGO sheet was filled with water. The vacuum chamber on the other side of the membrane was 40 Å along the z-direction (Supplementary Fig. 23b). A pressure of 150 MPa was applied to the graphene sheet to facilitate the water transport through the laminates.

All MD simulations were performed under the NVT ensemble using the GROMACS package[50]. A visual MD (VMD) program was used to visualize the trajectories[51]. The general AMBER force field (GAFF) topologies for the GO nanosheets, graphene nanosheets, and Glc molecules were generated using Sobtop code[52]. The GAFF has been widely employed in carbon-based systems and organic molecules[53]. In the current simulations, the laminar structure of GO was maintained. All the sp$^2$ carbon atoms in GO and graphene were treated as uncharged atoms. The charges of the functional groups of GO were adopted from ref. [54]. A large dihedral torsion barrier was applied to maintain the rigidity of the graphene sheet. The bond and angle parameters of the Glc molecule were generated based on the Hessian matrix, which was obtained at the B3LYP-D3(BJ)/def2-TZVP level of theory using ORCA[55]. The restrained electrostatic potential (RESP) atomic charges[56] of the Glc molecule were calculated using the Multiwfn program[57]. The SPC/E model was employed for the water molecules. The initial velocity of the system was randomly generated. The time step was 1 fs. Each system was simulated for 30 ns to investigate the water transport through the laminates. Furthermore, each system was simulated for 50 ns to investigate the stability of the laminate structures. The temperature was maintained at 300 K by a V-rescaled thermostat. The hydrogen-containing bonds were constrained using the LINCS algorithm[58]. The standard Lorentz-Berthelot mixing rule was used for the L-J interactions between the two particles. The particle mesh Ewald (PME) method[59] was employed to calculate long-range electrostatic interactions.

To better understand the interaction between the Glc molecule and rGO membrane in the confined laminate channel, a cluster model was cut from the last frame of the MD simulation trajectory to identify the nature of the interaction by applying the independent gradient model (IGM) method[60]. The edge carbons of the rGO sheets were saturated with hydrogen atoms. The IGM method was widely employed to identify and visualize intermolecular interactions using Multiwfn software[57]. The 2D scatter plot and the isosurface sign($\lambda_2$)$\rho$ mapped intermolecular electron density gradient difference ($\delta g\_inter$) for the Glc-in-rGO channel model were constructed to reveal the interactions.

## Data availability

Source data are provided with this paper.

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

## Acknowledgements

This work was supported by the Kobe University Strategic International Collaborative Research Grant Type B, Fostering Joint Research (H.M.), the Japan Society for the Promotion of Science KAKENHI, grant No. JP22H01849 (K.N.), and the National Natural Science Foundation of China, grant Nos. 22211540008, 22038006, and 21921006 (W.J.).

## Author contributions

K.G., W.J., and H.M. designed the research; K.G., Y.G., Z.L., Y.J., and Q.S. performed the research; K.G., K.N., T.Y., G.L., W.J., and H.M. analyzed the data; and K.G., Y.G., G.L., W.J., and H.M. wrote the paper.

## Competing interests

The authors declare no competing interests.
