## [Peer Review File · Nature Communications]

Deformation constraints of graphene oxide nanochannels under reverse osmosisREVIEWER COMMENTS

Reviewer #1 (Remarks to the Author):

In this manuscript, the authors investigated the channel deformation behavior of GO membranes by varying the channel chemistry and molecular intercalation for reverse osmosis of salt solutions. They show that the interlayer channel size of the modified GO membranes is dependent on the reaction time of reduction as well as the intercalated amount of Glc molecules. While the characterization of the Glc-rGO membranes and MD simulation results can help to analyze the effect of Glc molecules on the removal of ions as well as the stability of the Glc-rGO membranes during reverse osmosis processes, there are several questions to be clearly addressed with respect to the membrane structure and ion removal performance.

1. The representative XRD patterns of the Glc-rGO membranes are shown in the Figure 2e. Even though the pristine GO membrane exhibits the obvious peak in the XRD patterns, the modified GO membranes such as T40, T60, T80, Glc15 and Glc20 do not show the noticeable peaks. Thus, it is difficult to distinguish between noise and peaks, possibly indicating that interlayer channels are not appropriately formed. Supplementary evidence which shows the formation of nanochannels is required.

2. In Figure 2e, the peak in the XRD patterns shifted to higher 2-theta values with increase in the reaction time until T40, while the peak of T60 membrane shifted to lower 2-theta values. Although the same amount of Glc molecules is used to prepare the "y" membrane, why does the peak of the T60 membrane shift to lower 2-theta values? In addition, even though more Glc molecules are used to fabricate the Glc15 membrane, the peak position of the Glc15 membrane is similar to that of other Glc membranes. Additional discussion is required regarding the peak position of the Glc membranes.

3. In Figure 3b, the stability of the laminated GO structure is evaluated by Molecular dynamic simulations. On page 10, the authors mentioned that "however, when six Glc molecules were intercalated, the stability was improved". However, no distinct interaction (such as pi-pi interaction) between GO layers and Glc molecules is observed. It is necessary to clearly explain why the stability of the Glc-rGO membrane is improved, along with the reason of such "optimum point".

4. The channel size of the T10 membrane becomes broader in aqueous solution due to water intercalation, as shown in Figure 3c. However, the interlayer spacing of the T20 membrane is stably maintained under wet conditions. The authors claimed that the balance between the hydrophobicity of rGO and hydrophilicity of inserted Glc is important to constrain the deformation of the nanochannel. As can be seen in Figure 2d and Figure S6, there still are oxygen functional groups in the T20 membrane, but the membrane swelling is effectively suppressed. The authors should provide additional evidence to support their claims.

5. As Glc molecules have hydrophilic properties, the intercalated Glc molecules can be dissolved out during the permeation test. To confirm the inserted amount of the Glc molecules in the nanochannel upon the completion of the permeation test, additional characterization is required to quantify the leached amount of Glc molecules.

6. In Figure S10, after the first compaction process, the pristine GO membrane cannot sieve NaCl, resulting in almost 0% rejection rate. However, the pristine GO membrane exhibits 40% of the NaCl rejection rate after the secondary compaction process. It implies that a certain change occurs as a result of the successive compaction processes. If the laminated GO structure is successfully fabricated, the GO membrane should show the adequate NaCl rejection rate to some extent after the first compaction process. Please provide the reason why the pristine GO membrane fails to remove NaCl after the first compaction process?

7. As mentioned on page 13, the Glc10T60 has an optimal structure in terms of the inserted Glc ratio as well as the reaction time. Therefore, the NaCl rejection rate of the Glc10T60 stably shows above 90% during the filtration test (Figure 5c). However, the water permeance (Flux/Pressure) is

quite low for RO processes. In the RO process, the membranes should exhibit an excellent ion rejection rate and water flux performance at the same time. Thus, it is better to present a performance table for the comparison with other GO-based RO membranes.

8. On page 17, the authors mentioned "From the results of flux decline at high pressures (Figure 5), the Glc10T20 membrane exhibited the optimal performance, as it achieved moderate deformation constraints and limited intercalation of Glc molecules inside the nanochannels". However, it is insufficient to explain the degree of channel deformation from the results of the flux decline and the change in MWCO after the pressurization process. Additional evidence is required to confirm the structural change and to quantify the inserted amount of Glc molecules.

9. Finally, the purpose of this study is quite interesting, while it merely presents their experiments and results. However, considering the publication in Nature Communications, it would be a more influential study if the authors can reinforce their own scientific analysis and discussion, also providing any design perspective of the membrane based on their results.

Reviewer #2 (Remarks to the Author):

The manuscript investigates the behavior of layered graphene oxide membranes under pressurized conditions. The work systematically analyzes the performance of different graphene membranes under pressure and combines experimental results and molecular simulations. Although the work is well-conducted, the conclusions appear specific to the system investigated, and the results do not reveal generalizable new transport phenomena or promising membrane performance. My comments are further expanded on below.

The performance, even of the optimal membrane is quite poor, with water permeabilities 1-2 orders of magnitude lower than commercial RO membranes and salt permeabilities two orders of magnitude higher. It is unclear how the insights of this work will contribute to the development of improved membranes since performance of even the optimal membrane is orders of magnitude behind conventional polymers. Although the results give helpful insights into this specific system, it is unclear how they would apply to other nanofluidic processes. Membrane compaction effects are already well-documented in polymeric membranes and the studied effects seem somewhat comparable.

The mechanisms shown in Figure 1 and 3 (also described in line 143) where water influx occurs and water squeezes out seem speculative and it is unclear how they are supported by the data. From the explanation of the authors, it seemed like the membranes are perturbed at high pressures (and become more permeable and less selective) before stabilizing.

It is unclear how the properties of the Glc10T20 membrane were improved after deformation (line 291). The data appears to show that rejection and molecular weight cutoff remained constant.

The paper frequently refers to the membrane system as nanochannels. I am not aware of the conventions in this field, but the system appears to be an extremely heterogeneous network of angstrom-scale voids between the sheets.

Figure 1a is difficult to interpret. The left schematic shows osmotic pressure and applied pressure simultaneously but it is unclear whether the authors are presenting the schematic at equilibrium conditions (which is when this would apply). The meaning of the triangle in the nano channel is also unclear. Figure 2a may also be difficult for the reader to interpret. I would have assumed that the more reduced sample should have less oxygenated groups.

It is unclear how the slip flow described in the introduction is relevant for the system described.

Reviewer #3 (Remarks to the Author):

This paper is very important for understanding the microstructure evolution and filtration efficiency of assembled membrane of two-dimensional materials under pressure, and the authors need to clarify the following issues before publication.

1. In the figure3(b), the diagram should show how the pressure is applied. As shown in the figure S8, the structural are not force-balanced, how is the GO film supported without overall moving under the action of pressure? Boundary constraints for GO in MD should be clarified in the schematics and methods part.
2. In the line129-131, the authors suggest that Glc intercalation effect nanosheet stacking. Could the authors explain with more clear description how the intercalated Glc affect the evolution of the peak shift with the reaction time.
3. In the figure2(e) and figure3(c), for a better understanding, both the angle and corresponding interlayer distance could be present.
4. As the description in the lines 252-257(pages 14-15), in the initial pressurization process, the permeation flux first increases and then decreases, finally the permeation flux tends to be stable in the subsequent pressure cycle. (1) Is there any irreversible deformation of the GO channel occurring in this process, and what exactly is the deformation. (2) As the number of pressure cycles increases, will the permeation flux and selectivity of the GO remain stable? Will the deformation change periodically under pressure cycles?
5. In the lines 291-301(page 17), the results indicate that pressurization can both improve and deteriorate deformable nanochannel properties. For the Glc10T20 the separation performance is improved, while for the GO- and Glc20T60 membrane the performance is disrupted. The authors explain this may be resulted by the defects and disordered manner. Is there any randomness in the behavior of different membranes? Why do some of the points in the Figure5(e-f) have error bars and some don't?

Reviewer #4 (Remarks to the Author):

Multilayered graphene oxide membranes have been intensively studied in recent years but are still faced with big challenges in reverse osmosis. This manuscript provided new insights and perspectives of confinement states of graphene oxide interlayer nanochannels and demonstrated good results in effectively constraining the nanochannel robustness. The inconsistency of nanochannel confinement was evidenced by both experiments and simulations and valid desalination performance was obtained. Overall, the manuscript is well organized and could be publishable after minor revision. Below please find some detailed comments:

1. Regarding the construction of simulation models for different GO laminates, how did the authors decide the GO parameters (O/C ratio, interlayer spacing, etc.) and the number of glucose molecules in the nanochannel?
2. For desalination tests in Fig. 4b, how did the authors obtain the equilibrated results of water flux and salt rejection performance?
3. In Fig. 5e and 5f, why selecting glycerol, glucose, sucrose, and raffinose as solutes for rejection tests?
4. It is suggested to calculate and summarize the interlayer spacing values of interlayer nanochannels under different conditions in this paper for clear reference.

Responses to Reviewers' Comments

Manuscript ID: NCOMMS-22-31241

Manuscript type: Article

Title: Deformation constraint of graphene oxide nanochannels under reverse osmosis

Corresponding author: Hideto Matsuyama; Wanqin Jin

We are grateful to the reviewers for their insightful comments on our paper, which are extremely helpful in improving the quality of the manuscript. We have been able to incorporate changes to reflect most of the suggestions provided by the reviewers. The changes within the manuscript are highlighted in blue.

Please find below a point-by-point response to the reviewers' comments and concerns.

Comments from Referee 1

- **General Comment:**

In this manuscript, the authors investigated the channel deformation behavior of GO membranes by varying the channel chemistry and molecular intercalation for reverse osmosis of salt solutions. They show that the interlayer channel size of the modified GO membranes is dependent on the reaction time of reduction as well as the intercalated amount of Glc molecules. While the characterization of the Glc-rGO membranes and MD simulation results can help to analyze the effect of Glc molecules on the removal of ions as well as the stability of the Glc-rGO membranes during reverse osmosis processes, there are several questions to be clearly addressed with respect to the membrane structure and ion removal performance.

Response:

Thanks for the reviewer's evaluation. We have revised manuscript according to the reviewer's valuable comments. The changes within the manuscript are highlighted in blue. Please find below a point-by-point response to the comments and concerns.

- **Comment 1:**

The representative XRD patterns of the Glc-rGO membranes are shown in the Figure 2e. Even though the pristine GO membrane exhibits the obvious peak in the XRD patterns, the modified GO membranes such as T40, T60, T80, Glc15 and Glc20 do not show the noticeable peaks. Thus, it is difficult to distinguish between noise and peaks, possibly indicating that interlayer channels are not appropriately formed. Supplementary evidence which shows the formation of nanochannels is required.

Response:

Thanks for the reviewer's constructive comments. The characteristic peak intensity of reduced GO membranes was relatively low, which is due to the loss of oxygenated functional groups compromising the crystalline peak of GO and thin laminate layer

deposited on substrate. To better evidence the formation of layered structure and nanochannel, we have prepared membrane samples with higher nanosheet loading by depositing 5-fold amount of GO or rGO for XRD analysis. From the XRD patterns of thicker samples, the characteristic peaks were clearer to distinguish, indicating the existence of layered structure and interlayered nanochannel. In addition, the tendency of the peak shift in thicker samples was similar to that in the normal ones (peak of T“y” membrane shifting to higher 2θ value first and then to lower one; peak of Glc“x” membrane shifting to higher 2θ value first and then to lower one and finally kept in similar positions). Therefore, the nanochannel formation and the peak shifting results of XRD were confirmed.

We have included the XRD results of membrane samples with higher nanosheet loadings in revised manuscript and Supplementary Information:

Manuscript (RESULTS – Controlling GO membrane nanochannels)

*“...The position shift of the GO characteristic peaks (d_{001} reflection) indicated in the XRD spectra was **inconsistent** with the change in reaction conditions (**Figure 2e and Supplementary Table 1**). With **an** increase in the reaction **period**, the peak shifted to higher 2θ values until T40 and then to lower 2θ values for T60 (**the** peak intensity of T80 is **too** low to assign). With **an** increasing Glc ratio, the peak position did not significantly change from Glc1 to Glc15 (**the** peak intensity of Glc20 was **too** low to assign). **Clearer** characteristic peaks could be observed in laminate samples with increased deposition amount, and similar peak shifting tendencies with a change in the reaction conditions were confirmed (**Supplementary Figure 8**). These results tentatively reveal the Glc intercalation effect during nanosheet stacking...”*

Supplementary Information

“*Supplementary Fig. 8. XRD spectra of thicker GO and Glc-rGO samples (dashed lines and arrows indicate peak shifts). These thicker samples were prepared by using 5-fold GO or Glc-rGO deposition amounts of normal samples.*”

• **Comment 2:**

In Figure 2e, the peak in the XRD patterns shifted to higher 2-theta values with increase in the reaction time until T40, while the peak of T60 membrane shifted to lower 2-theta values. Although the same amount of Glc molecules is used to prepare the T“y” membrane, why does the peak of the T60 membrane shift to lower 2-theta values? In addition, even though more Glc molecules are used to fabricate the Glc15 membrane, the peak position of the Glc15 membrane is similar to that of other Glc membranes. Additional discussion is required regarding the peak position of the Glc membranes.

Response:

Thanks for the reviewer’s question and comment. The amount of Glc used for suspension preparation was not the final amount of Glc retained in the membrane nanochannels, as some of Glc molecules was filtrated out during membrane preparation.

While same amount of Glc molecules was used for T“y” membrane, different reaction time induced different reduction degree of GO, which might affect the interactions between Glc and nanosheets and nanosheet stacking order. For T60

membrane, higher reduction degree of GO might induce worse compatibility with Glc molecules causing unfavorable stacking during membrane preparation and enlarging the overall interlayer spacing.

For Glc“x” membranes, on the one hand, higher amount of Glc used would lead to higher reduction of GO, decreasing the interlayer spacing; on the other hand, larger amount of Glc might also have higher possibility to be retained in laminates with a larger amount, increasing the interlayer spacing. As a result, the interlayer spacing change might be restricted by this trade-off, keeping a similar peak position as revealed in XRD spectra.

We have also performed experiments to quantify the finally intercalated amount of Glc molecules in different membranes to reveal the factors (Glc intercalation or unfavorable nanosheet stacking) resulting in the interlayer changes, according to this reviewers' other comments (see Response to **Comment 8**).

We have included additional discussion in the context of the manuscript:

Manuscript (RESULTS – Controlling GO membrane nanochannels)

“These results tentatively reveal the Glc intercalation effect during nanosheet stacking, as a higher reduction degree is typically accompanied by a continuous peak shift to higher 2θ values¹⁹. Although the same amount of Glc molecules was used for T’y’ membrane, different reaction periods induced different reduction degree of GO, which affected the interactions between the Glc and the nanosheets and nanosheet stacking order. A higher reduction degree of GO may induce worse compatibility with the Glc molecules, causing unfavorable stacking during filtration-induced assembly and increasing the overall interlayer spacing. A higher amount of Glc used would lead to a higher reduction of GO in the Glc‘x’ membranes, decreasing the interlayer spacing. However, higher amount of Glc may also have a higher possibility of retaining more Glc molecules, increasing the interlayer spacing. Consequently, this trade-off may restrict the interlayer spacing change, maintaining a similar peak position, as

shown in the XRD spectra. The shift in the peak position was limited to the narrow range of 11–12°...

- **Comment 3:**

In Figure 3b, the stability of the laminated GO structure is evaluated by Molecular dynamic simulations. On page 10, the authors mentioned that “however, when six Glc molecules were intercalated, the stability was improved”. However, no distinct interaction (such as pi-pi interaction) between GO layers and Glc molecules is observed. It is necessary to clearly explain why the stability of the Glc-rGO membrane is improved, along with the reason of such “optimum point”.

Response:

Thanks for the comment. In order to get a better understanding to the interaction between Glc molecule and rGO membrane in the confined laminate channel, a cluster model was used to identify the nature of the interaction between Glc and rGO. Results revealed weak attractive van der Waals interaction and relatively strong attractive interaction, including hydrogen bond. With the increased Glc amount in laminates, more favorably interactive sites would be involved which results in enhanced stability from Glc(2)-rGO to Glc(6)-rGO model.

From experimental results, with certain Glc amount used for suspension preparation, better stability could be observed (e.g., T20 membrane) which is in accordance with simulation results. However, in experiments, excessively high Glc amount in suspension resulted in worse stability of laminates (e.g., G20 membrane). Since with the increase of Glc concentration, the reduction degree of GO is also increased, leading to the decrease of the hydrogen bonds between Glc and rGO. Therefore, there should be a proper concentration of Glc that can stabilize the laminates as found in experimental work through the attractive hydrogen bond interaction.

We have included new simulation results and more discussion in the manuscript and Supplementary Information:

Manuscript (RESULTS – Nanochannel confinement in water)

“Molecular dynamics (MD) simulations were performed to *investigate* the evolution of the laminate structure under water pressure conditions *and to reveal the contribution of Glc to the structural stability* (**Figure 3a**, **Supplementary Figure 9**, and **Supplementary Movies 1–4**). rGO laminates with lower O/C ratios *exhibited* a more stable structure during a simulation of 50 ns, compared to GO, *which is ascribed to the increased hydrophobicity of rGO*. *When two Glc molecules were intercalated into each channel of the rGO laminates, the laminate structural stability was negligibly affected. However, when six Glc molecules were intercalated, the stability was improved. To identify the nature of the interaction between the Glc and rGO, a cluster model (Supplementary Figure 10) from the last frame of the MD simulation trajectory was employed by applying the independent gradient model (IGM) method. The 2D scatter plot and the isosurface ($\text{sign}(\lambda_2)\rho$ mapped intermolecular electron density gradient difference (δg_{inter})) for the model are shown in **Figure 3b**. The region close to $\text{sign}(\lambda_2)\rho = 0$ (green) denotes mainly the weak attractive van der Waals interactions. The cyan-to-blue region denotes relatively strong attractive interactions, including hydrogen bonds (**Figure 3c**). Thus, increasing the Glc amount in certain GO laminates stabilizes them to a certain extent. However, for practical Glc-rGO laminate preparation, the increase in Glc concentration was accompanied by an increased degree of reduction in GO, decreasing the number of hydrogen bonds between Glc and rGO. Therefore, an appropriate concentration of Glc should stabilize the membrane through attractive hydrogen bonds and van der Waals interactions.”*

“Figure 3. **Nanochannel deformation in water.** (a) MD simulations of laminate models of GO, rGO, and Glc-rGO in pressurized water. Pressures of 50 and 10 MPa were applied on the opposite sides of the simulation cell for all the models, as indicated in the GO model as an example. Two Glc-rGO models with two and six Glc molecules in each channel were constructed, denoted as Glc(2)-rGO and Glc(6)-rGO, respectively; (b) $\text{sign}(\lambda_2)\rho$ mapped 2D scatter plot of the intermolecular electron density gradient difference (δg_{inter}). The value of $\text{sign}(\lambda_2)\rho$ is represented by filling color according to the color bar. ρ denotes the electron density at the weak interaction critical point, $\text{sign}(\lambda_2)$ denotes the sign of the second largest eigenvalue λ_2 of the electron density Hessian matrix; (c) Isosurface of δg_{inter} with an isovalue of 0.005 a.u.. The blue isosurface between the oxygen-containing group of rGO and the hydroxyl group of Glc indicates the C-H...O hydrogen bond interactions as depicted by dashed lines; (d) XRD spectra of wetted GO and Glc-rGO samples (dashed lines and arrows indicate peak shifts); (e) Time-course pure water flux of GO and Glc-rGO samples at 1 MPa. Glc ratio for Glc-rGO samples derived from different reaction periods (T0–T80) was fixed at 10 (Glc10), whereas the reaction period for Glc-rGO samples derived from different Glc ratios (Glc0–Glc20) was fixed at 60 min (T60).”

Manuscript (METHODS – Computational simulation details)

“To better understand the interaction between the Glc molecule and rGO membrane in the confined laminate channel, a cluster model was cut from the last frame of the MD simulation trajectory to identify the nature of the interaction by applying the independent gradient model (IGM) method⁶⁰. The edge carbons of the rGO sheets were saturated with hydrogen atoms. The IGM method was widely employed to identify and visualize intermolecular interactions using Multiwfn software⁵⁷. The 2D scatter plot and the isosurface $\text{sign}(\lambda_2)\rho$ mapped intermolecular electron density gradient difference (δg_{inter}) for the Glc-in-rGO channel model were constructed to reveal the interactions.”

Supplementary Information

“**Supplementary Fig. 10.** Glc-in-rGO channel model from the last frame of the MD simulation trajectory used for the IGM analysis.”

• **Comment 4:**

The channel size of the T10 membrane becomes broader in aqueous solution due to water intercalation, as shown in Figure 3c. However, the interlayer spacing of the T20 membrane is stably maintained under wet conditions. The authors claimed that the balance between the hydrophobicity of rGO and hydrophilicity of inserted Glc is important to constrain the deformation of the nanochannel. As can be seen in Figure 2d and Figure S6, there still are oxygen functional groups in the T20 membrane, but

the membrane swelling is effectively suppressed. The authors should provide additional evidence to support their claims.

Response:

As suggested, wetting properties of different laminate samples were characterized. The improved anti-swelling property of T20 membrane was due to the reduced oxygen functional groups and accordingly enhanced hydrophobicity as compared with T10. Compared with original GO, T10 showed even higher O/C ratio due to introduced Glc while T20 showed obviously decreased ratio due to significant chemical reduction. Therefore, T20 exhibited enhanced hydrophobicity and it became more difficult for water to wet and penetrate the T20 membrane. The under-layered laminates of T20 thus could be kept from wetting, showing maintained interlayer spacing under wet conditions. However, as the reviewer mentioned, there still were retained oxygen functional groups in T20 membrane, and this explains why the T20 interlayer could be wetted under more harsh conditions (e.g., wetted by NaCl solution, Figure 4a).

We characterized the wetting properties of all membrane samples by measuring water contact angle in air. The water contact angle of T20 membrane was significantly increased in comparison with that of GO and T10 membrane, indicating the enhanced hydrophobicity that could improve the resistance of water wetting.

We have added above characterization results and discussion in the manuscript and Supplementary Information:

Manuscript (RESULTS – Nanochannel confinement in water)

*“...The GO, T0, and T10 laminates shifted to lower 2θ values of $\sim 7^\circ$ from the original position of $11\text{--}12^\circ$ (**Figure 2e**), indicating significant water intercalation into the interlayered channels. Better water wetting properties (lower water contact angles) were obtained for the GO, T0, and T10 laminates than for the other samples (**Supplementary Figure 11**). Hence, water penetrates easier in the interlayers of these laminates, increasing the interlayer spacing. For other samples exhibiting higher water contact angles, a similar XRD peak position was maintained as that under dry*

conditions, but the peak shifted with an increased reaction period (T60 or higher) and Glc ratio (Glc10 or higher). This indicates that the Glc molecules included in the nanochannel may also have an impact. Because the nanochannel is ...”

Supplementary Information

“**Supplementary Fig. 11.** Water contact angle in air on the surfaces of GO and different Glc-rGO membranes.”

- **Comment 5:**

As Glc molecules have hydrophilic properties, the intercalated Glc molecules can be dissolved out during the permeation test. To confirm the inserted amount of the Glc molecules in the nanochannel upon the completion of the permeation test, additional characterization is required to quantify the leached amount of Glc molecules.

Response:

As suggested, we have performed experiments to quantify the leached amount of Glc molecules from membranes after permeation test. In the quantification experiment, fixed amount of feed ultrapure water was used and the retentate and permeate were

reflowed to the feed solution. After 24 h permeation test, the solution was subjected to total organic carbon analysis to measure the leached Glc amount. It was found that the leached amount of Glc is with the order of Glc10T20 < Glc10T60 < Glc20T60. It corresponds well with flux decline order of these membranes, indicating more Glc would leach for laminates with lower structural stability. Nevertheless, the leached Glc for these membranes only took a limited portion (e.g., 5% for Glc10T20) of the total content in the laminates. We have included the results in revised manuscript and Supplementary Information:

Manuscript (RESULTS – Role of the deformation constraint of nanochannel for salt/water separation)

“The improved hydrophobicity of these three Glc-rGO membranes (Supplementary Fig. 11) indicates the improved water stability...

As Glc molecules have hydrophilic properties, the intercalated Glc molecules can also be dissolved with pressurized flow during the permeation test (Figure 5f). The leached amounts of Glc for these membranes during permeation tests were in the order of Glc10T20 < Glc10T60 < Glc20T60 (5%, 15%, and 24%, respectively; Figure 5e). This order is also consistent with their flux decline tendency, indicating that the presence of Glc molecules in the nanochannels was influenced by channel stability. Higher Glc leaching indicates lower integration of Glc and rGO and, thus, a higher possibility of a perturbed structure during Glc leaching by pressurized flow...”

“Figure 5. **Nanochannel deformation and performance in pressure-changing reverse osmosis.** Filtration of NaCl solutions with pressure variations by (a) GO, (b) Glc10T20, (c) Glc10T60, and (d) Glc20T60; (e) Glc content in original membrane precursor suspension, intercalated Glc content in prepared membranes, and leached Glc content from membranes during permeation test. The error bars represent the standard deviations of three measurements; (f) Illustration of Glc leaching from laminates with the pressurized flow; (g) Molecular weight cut-off curves of membranes before and after pressure variations. The error bars represent the standard deviations of three measurements.”

Supplementary Information

“Supplementary Fig. 17. Schematic diagram of apparatus setups for measuring the (a) intercalated Glc amount in stacked laminates and (b) leached Glc amount from the membrane during the permeation test.”

“To measure the amount of Glc intercalated in the stacked laminates, the filtrate was collected after vacuum filtration of the suspension with a known amount of added Glc for the preparation of the laminates. The filtrate was analyzed using the total organic carbon (TOC) technique to quantify the Glc. Hence, the amount of Glc intercalated in the laminates was obtained. Control samples of the pristine GO suspension (without Glc addition) were also prepared to exclude the TOC from the filtered impurities of materials other than Glc.

Ultrapure water with a known volume was used as the feed to measure the amount of leached Glc during the permeation test. Pressures of 1 and 2 MPa were successively applied for the permeation test with a duration of 12 h for each pressure condition, and the retentate and permeate solutions were reintroduced to the feed tank. After the permeation test, the TOC of the solution was measured to determine the amount of Glc leached from the membrane. Control permeation tests for pristine GO membranes were also performed to exclude the leached TOC from materials other than Glc in the membranes.”

- **Comment 6:**

In Figure S10, after the first compaction process, the pristine GO membrane cannot sieve NaCl, resulting in almost 0% rejection rate. However, the pristine GO membrane exhibits 40% of the NaCl rejection rate after the secondary compaction process. It implies that a certain change occurs as a result of the successive compaction processes. If the laminated GO structure is successfully fabricated, the GO membrane should show the adequate NaCl rejection rate to some extent after the first compaction process. Please provide the reason why the pristine GO membrane fails to remove NaCl after the first compaction process?

Response:

Thanks for the comment and question. Pristine GO is quite hydrophilic and easily swelled by water. The duration of first compaction process was too short to make GO membrane completely compacted, as the water flux of GO membrane continued to decrease at a relatively high rate. While the GO membrane was not competently compacted, feed changing from water to NaCl solution after first compaction made the serious swelling of GO membrane (a more than 20-fold flux increase), resulting in almost zero rejection. Therefore, in the second compaction process, we prolonged the duration until the flux decline of GO membrane was minimized. In this case, GO membrane was much more compacted than the one after the first compaction process. As observed, feed changing from water to NaCl solution after the second compaction process did not make significant swelling of the GO membrane (a slight flux increase). Hence, the GO membrane after this sufficient compaction could show moderate rejection towards NaCl.

We have added above discussion in the Supplementary Information:

“**Supplementary Fig. 13.** Filtration performance of the pristine GO membrane with different pre-compaction periods. Filtrations of pure water and NaCl solution (500 ppm) were conducted under 1 MPa. Membranes typically need to be compacted before use. Here, after compaction for approximately 100 min by pure water filtration (first compaction), the pristine GO membrane still exhibited significant instability towards the filtration of the NaCl solution, which was revealed by the gradual decline in permeation flux and the nearly 0% rejection. Therefore, we extended the compaction period to approximately 16 h (second compaction), after which we observed that the NaCl rejection was increased to >40%. Thus, the pre-compaction period of 16 h was applied for all the membranes before any other following performance evaluations.

The duration of the first compaction process was too short to completely compact the GO membrane as the water flux of the GO membrane continued to decrease at a high rate. While the GO membrane was not completely compacted, changing the feed water to NaCl solution after the first compaction caused severe swelling of the GO membrane (a more than 20-fold flux increase), resulting a nearly 0% rejection. Therefore, in the second compaction process, the duration was prolonged until the flux decline of the GO membrane was minimized. In this case, the GO membrane was significantly more compacted than that after the first compaction process. It can be observed that changing the feed water to the NaCl solution after the second compaction process did not cause significant swelling of the GO membrane (a slight flux increase). Hence, the GO membrane after this sufficient compaction, the membrane exhibited moderate rejection of NaCl.”

- **Comment 7:**

As mentioned on page 13, the Glc10T60 has an optimal structure in terms of the inserted Glc ratio as well as the reaction time. Therefore, the NaCl rejection rate of the Glc10T60 stably shows above 90% during the filtration test (Figure 5c). However, the water permeance (Flux/Pressure) is quite low for RO processes. In the RO process, the membranes should exhibit an excellent ion rejection rate and water flux performance at the same time. Thus, it is better to present a performance table for the comparison with other GO-based RO membranes.

Response:

We agree with the reviewer's comments. As we know, the studies on two-dimensional-material membranes just begun in recent years, and there are still a lot of fundamental challenges and technical issues to be addressed for structural optimization and mechanistic understandings. The number of publications reporting GO membranes for NaCl rejection is quite limited compared to the separation of other relatively larger solutes, which is also the motivation for us to conduct this study. This study focused on the mechanism of membrane nanochannel inconsistency during membrane process. The insights into the mechanism can help to optimize membrane structure and improve membrane performance. Realizing this process may take a long time, which is also similar to the case of polyamide RO membranes, as the current performance of polyamide membranes was achieved through intensive efforts over more than half century. We have included above discussion and provided a performance list of state-of-the-art GO-based membranes for NaCl rejection and included it in the revised manuscript and Supplementary Information.

Manuscript (RESULTS – Role of the deformation constraint of nanochannel for salt/water separation)

“...Long-term performance evaluation (>150 d) at 3 MPa was conducted using various NaCl feed concentrations ranging from 500 to 5000 ppm (Figure 6c). Each feed concentration was tested for approximately one month, and the membrane exhibited stable NaCl rejection throughout the entire period. The membrane flux

gradually decreased because of the increased osmotic pressure at high NaCl concentrations *and* possible membrane compaction. However, the optimal membrane, possessing a strong constraint on nanochannel deformation, demonstrated promising results. While this study aims to investigate the nanochannel deformation using Glc, possessing no distinct interactions with the GO, the obtained stable NaCl rejection performance is still comparable to studies on GO-based membranes (**Supplementary Table 3**). The findings of this study could aid in designing modifications and intercalations for GO membranes to achieve nanochannel deformation constraints and satisfactory flux and salt rejection. For example, grafting agents that possess distinct chemical structures and functional groups (such as π -conjugated molecules with water-affinity functional groups) can stabilize the nanochannel and favor water diffusion, which should work better than the weakly interacted Glc demonstrated in this study.”

Supplementary Information

“**Supplementary Table 3.** Summary of literature reports on GO-based membranes exhibiting relatively high NaCl rejection.”

Membrane code	Permeance [L m ⁻² h ⁻¹ bar ⁻¹]	Feed NaCl concentration [ppm]	NaCl rejection [%]	Reference
PE@ArGO	2.9	58.4	88	2
GO-FLG	0.35	2000	85	3
GO (EPR)	25	500	97	4
GNM/SWNT	110.6	2000	85	4
TU-GO	1.5	100	95	5
T CPP-rGO	0.17	500–2000	85–92	6
Air-rGO	1.05	1000	83	7
K-rGO	0.6	1000	91	8
GO-TBO	0.4	584–29220	30–85	9
TA-rGO	0.36	500–2000	80–92	11
GO/g-C ₃ N ₄	33.5	1170	87	12
GO-AQP	7.83	1000	99.1	13
Glc-rGO (This study)	0.14	500–5000	88.7–93.2	

- **Comment 8:**

On page 17, the authors mentioned “From the results of flux decline at high pressures (Figure 5), the Glc10T20 membrane exhibited the optimal performance, as it achieved moderate deformation constraints and limited intercalation of Glc molecules inside the nanochannels (Figure 5)”. However, it is insufficient to explain the degree of channel deformation from the results of the flux decline and the change in MWCO after the pressurization process. Additional evidence is required to confirm the structural change and to quantify the inserted amount of Glc molecules.

Response:

As suggested, we have provided more evidence to confirm the structural change using XRD analysis and calculated pore radius distribution. XRD analysis was used to reveal the overall interlayer spacing change before and after pressurized filtrations for GO, Glc10T20, Glc10T60, and Glc20T60 laminates (peak intensity of Glc20T60 is too weak to assign). The results also confirmed that the structure of GO was disrupted after filtration while those of Glc10T20 and Glc10T60 were more densely packed with a narrowed interlayer spacing. The pore size distribution was also possible to calculate from the MWCO plot for GO and Glc10T20 laminates to indicate the change of pore size (parameters needed for calculation are not available for MWCO curves of Glc10T60 and Glc20T60). In addition, we characterized the surface morphologies of these samples before and after filtrations to confirm that while these membranes kept intact after pressurization, the internal laminate structure was altered, with changed overall interlayer spacing as well as rejection capacity to salt and neutral organic solutes. The inserted amount of Glc molecules was also quantified and the result is consistent with the assumption and the observed results from other analysis and evaluations.

Below lists the revisions we made in the manuscript and Supplementary Information:

Manuscript (DISCUSSION – Role of the deformation constraint of nanochannel for salt/water separation)

“The Glc10T60 membrane with optimal nanochannel stability achieved the highest and most stable NaCl rejection (~90%) during filtration with pressure variation, although its flux decline at increased pressures was higher than that of Glc10T20 (Figure 5c). However, the Glc20T60 membrane performed poorly with a NaCl rejection similar to that of Glc10T20 (~60%) but with a significantly lower permeation flux (Figure 5d), although these laminates remained intact after pressurized filtration (Supplementary Figure 15). The filtration performances of different membranes at the 1 MPa steps in the filtration cycle...

The improved hydrophobicity of these three Glc-rGO membranes (Supplementary Figure 11) indicates the improved water stability of their bulk structures. This was inconsistent with their nanochannel stability and salt rejection, as they had different responses to the infiltrated water flow in their different pore interiors. The amount of Glc retained after the preparation of these laminates could be different, considering the different degrees of GO reduction and interactions with Glc. The amount of Glc intercalated was similar for the Glc10T20 and Glc10T60 laminates, whereas it was higher for Glc20T60 (Figure 5e and Supplementary Figure 17), indicating that higher amounts of Glc used in suspension lead to a higher possibility of retained Glc in the corresponding laminates. Although this explains the difference in performance between Glc10 and Glc20, the difference between T20 and T60 must be further investigated.

...

The molecular weight cut-off (MWCO, defined as the lowest molecular weight solute in which the membrane retains 90% of the solute) of these membranes was investigated to reveal their separation capability toward neutral solutes and channel dimensions before and after the pressure variations. Rejection tests for organic solutes with a suitable molecular weight³⁶, including glycerol (92 Da), Glc (180 Da), sucrose (342 Da), and raffinose (504 Da), were conducted to obtain MWCO curves (Figure 5g). Before the pressure variation, the MWCO values of the different membranes were ~180–500 Da, and the order corresponded to the NaCl rejection results. After pressure variations, nanochannel deformation occurred as the MWCO values of GO, Glc10T20,

and Glc20T60 changed, whereas the optimal Glc10T60 remained stable (Supplementary Table 2). GO and Glc20T60 were transformed into less confined nanochannels, whereas Glc10T20 improved. The pore radius distributions of GO and Glc10T20 (Supplementary Figure 18) were attainable according to their MWCO curves, where the GO exhibited similar results after the pressurization, whereas Glc10T20 exhibited a narrower distribution and smaller pore radius. This indicates that pressurization can either improve or deteriorate the deformable nanochannel properties for separating molecules within a certain size range. From the results of flux decline at high pressures (Figure 5a–d), the Glc10T20 membrane exhibited the optimal performance, as it achieved moderate deformation constraints by limited intercalation and leaching of Glc molecules inside the nanochannels (Figure 5e). Hence, it was confirmed that pressurization can improve the packing of multilayered structures, and in some cases, improve GO nanochannel membranes to render higher separation performance.²² In contrast, the nanochannels of GO- and Glc20T60-contained nanosheets with a wider deformation range were sensitive to external conditions and likely to be disrupted in a disordered manner and form defects. The XRD spectra of these laminates before and after the pressurized filtration test thus were compared (Supplementary Figure 19). A significant decrease in the peak intensity was observed for pristine GO, whereas a similar intensity but smaller interlayer spacing (peak shifting to higher 2θ values) was observed for Glc10T20 and Glc20T60. The results and rejection performance toward NaCl and organic solutes further confirmed divergent structural changes in mutable GO-based laminates. To fabricate a stable nanochannel membrane...”

“Figure 5. **Nanochannel deformation and performance in pressure-changing reverse osmosis.** Filtration of NaCl solutions with pressure variations by (a) GO, (b) Glc10T20, (c) Glc10T60, and (d) Glc20T60; (e) Glc content in original membrane precursor suspension, intercalated Glc content in prepared membranes, and leached Glc content from membranes during permeation test. The error bars represent the standard deviations of three measurements; (f) Illustration of Glc leaching from laminates with the pressurized flow; (g) Molecular weight cut-off curves of membranes before and after pressure variations. The error bars represent the standard deviations of three measurements.”

Supplementary Information

“*Supplementary Fig. 15. SEM images with different magnifications exhibiting the surface morphologies of the GO, Glc10T20, Glc10T60, and Glc20T60 membranes before and after pressurized filtrations.*”

“*Supplementary Fig. 18. Fitted pore radius distribution of GO and Glc10T20 membranes.*”

“The pore radius distribution curve was expressed as a probability density function.¹ The mean pore radius of the membrane was assumed to be the Stokes radius of the organic solute with a measured rejection of 50%. The geometric standard deviation of the distribution is assumed to be the geometric standard deviation of the probability density function curve, which is the ratio of the Stokes radius with a rejection of 84.13% to that with a rejection of 50%. The pore radius distribution of the membrane can thus be expressed as:

$$\frac{dR(r_p)}{dr_p} = \frac{1}{r_p \ln \sigma_p \sqrt{2\pi}} \exp \left[-\frac{(\ln r_p - \ln \mu_p)^2}{2(\ln \sigma_p)^2} \right]$$

where r_p denotes the Stokes radius of the organic solute, σ_p the geometric standard deviation of the distribution curve, and μ_p the mean pore radius.

The Stokes radii of organic solutes can be calculated as

$$\ln(r_p) = -1.4962 + 0.4654 \ln(M_w)$$

where M_w denotes the molecular weight of the organic solute.”

“**Supplementary Fig. 19.** XRD spectra of the GO, Glc10T20, Glc10T60, and Glc20T60 laminates before and after the pressurized filtrations.”

“XRD spectra were obtained for the laminate samples before and after continuous pressurized water filtration under pressures of 1 and 2 MPa for a duration of 12 h for each pressure condition.”

- **Comment 9:**

Finally, the purpose of this study is quite interesting, while it merely presents their experiments and results. However, considering the publication in Nature Communications, it would be a more influential study if the authors can reinforce their own scientific analysis and discussion, also providing any design perspective of the membrane based on their results.

Response:

Thanks for the reviewer’s positive comment on the purpose of this study. According to the reviewer’s valuable comments, we have performed additional experiments to make further analysis for addressing the reviewer’s concerns. In addition, for generality of this work, we have added a Discussion section in the manuscript to provide some design perspectives on GO and similar two-dimensional-material (e.g., Ti₃C₂Tx, boron nitride, and carbon nitride) membranes for potential RO and other nanofluidic applications (e.g., salinity gradient power generation, flow reactor):

Manuscript (RESULTS – Role of the deformation constraint of nanochannel for salt/water separation)

*“...Long-term performance evaluation (>150 d) at 3 MPa was conducted using various NaCl feed concentrations ranging from 500 to 5000 ppm (**Figure 6c**). Each feed concentration was tested for approximately one month, and the membrane exhibited stable NaCl rejection throughout the entire *period*. The membrane flux gradually decreased because of the increased osmotic pressure at high NaCl concentrations *and* possible membrane compaction. However, the optimal membrane,*

possessing a strong constraint on nanochannel deformation, demonstrated promising results. While this study aims to investigate the nanochannel deformation using Glc, possessing no distinct interactions with the GO, the obtained stable NaCl rejection performance is still comparable to studies on GO-based membranes (Supplementary Table 3). The findings of this study could aid in designing modifications and intercalations for GO membranes to achieve nanochannel deformation constraints and satisfactory flux and salt rejection. For example, grafting agents that possess distinct chemical structures and functional groups (such as π -conjugated molecules with water-affinity functional groups) can stabilize the nanochannel and favor water diffusion, which should work better than the weakly interacted Glc demonstrated in this study.”

Manuscript (DISCUSSION)

“In summary, we investigated the nanochannel deformation behavior of multilayered GO membranes and determined the balance between the channel chemistry and confinement stability for the reverse osmosis of salt solutions. The deformation depended on the pore wall (GO nanosheets) and pore interior (intercalated moieties) of the nanochannel. Notably, while minor changes to the pore interior negligibly altered the channel dimensions from typically characterized values, they may result in a significant difference in the nanochannel stability (deformation constraint) during reverse osmosis. Thus, the discord between the characterization and performance of the GO nanochannel membranes was highlighted. The effect of the intercalation chemistry should be carefully considered when designing nanochannel membranes using functionalized nanosheets to construct confined nanochannels. There is still a considerable performance gap between the state-of-the-art GO-based membranes and commercial polyamide membranes for reverse osmosis. Therefore, future efforts are required to improve the quality of raw GO materials and search for effective modification agents. Suitable modification agents that can constrain the swelling or compaction of laminates and exhibit low transport resistance in nanochannels are

desirable for maintaining constrained nanochannels for precise ion separation. In addition, large-scale production of high-quality GO nanosheets is expected in future to make the fabrication of few-layered laminates more practical for maximizing water flux.

The phenomenon observed in the GO membrane nanochannels under reverse osmosis could also provide useful information for maintaining channel confinement for materials of other laminates and applications of other nanofluidic processes. Although the material and application investigated in this study are specific, we believe that the results on what affects channel confinement and how to maintain it are also applicable to other systems. For example, the nanochannel deformation and its constraint could be extended to laminate membranes, including MXene (Ti_3C_2Tx)^{37,38}, boron nitride^{39,40}, and carbon nitride⁴¹, where functionalization is necessary to achieve stable nanochannels during pressurized membrane separation processes in water, organic solvents, or gas. In addition to the membrane process, the factors affecting the channel confinement constraint are also relevant for other nanofluidic processes involving aqueous chemistry within nanometric slit pores⁶. For example, nanochannel confinement enables controllable ion transport in laminates, exhibiting significant potential for salinity gradient power generation⁴² in the water-energy nexus, and fluid confinement in laminates boosted radical yields for efficient chemical reactions⁴³ in advanced oxidation processes. However, it is recommended that the consistency of channel confinement in nanofluidic applications be evaluated by designing experiments based on the characteristics of specific processes, thereby acquiring incremental insights into the nanochannel deformation behavior of different materials and applications.”

Comments from Referee 2

General Comment:

The manuscript investigates the behavior of layered graphene oxide membranes under pressurized conditions. The work systematically analyzes the performance of different graphene membranes under pressure and combines experimental results and molecular simulations. Although the work is well-conducted, the conclusions appear specific to the system investigated, and the results do not reveal generalizable new transport phenomena or promising membrane performance. My comments are further expanded on below.

Response:

Thanks for the reviewer's evaluation and providing the valuable comments.

For “the conclusions appear specific to the system investigated”, GO is the most typical and mostly studied two-dimensional materials, the nanochannel properties found for GO are also applicable for others. While our study is focused on the specific materials (GO) for specific application (pressurized filtration), similar phenomenon or issues could also be revealed from other two-dimensional-material membranes (e.g., two-dimensional MXene requiring functionalization for structural stability) and from other applications (e.g., ion transport in osmotic energy harvesting). We have revised the manuscript to provide some perspectives on the findings in this work.

For “the results do not reveal generalizable new transport phenomena or promising membrane performance”, this study characterized the factors rendering the instability of laminates during pressurized filtrations and provided some insights into the dynamic laminate structure for better considerations of future membrane design. It was found that the commonly characterized nanochannel is not consistent when external conditions change. The change of the nanochannel will also affect the water and ion transport within it, highlighting the importance to include effects of inconsistency of nanochannels for reports on studied nanochannel transport phenomenon or obtained high membrane performance.

For membrane performance, the studies on two-dimensional-material membranes just begun in recent years, and there are still a lot of fundamental challenges and technical issues to be addressed for structural optimization and mechanistic understandings. The number of publications reporting GO membranes for NaCl rejection is quite limited compared to the separation of other relatively larger solutes, which is also the motivation for us to conduct this study. This study focused on the mechanism of membrane nanochannel inconsistency during membrane process. The insights into the mechanism can help to optimize membrane structure and improve membrane performance.

Revisions have been included in the revised manuscript according to the reviewer's valuable comments. The changes within the manuscript are highlighted in blue. Please find below a point-by-point response to the comments and concerns.

- **Comment 1:**

The performance, even of the optimal membrane is quite poor, with water permeabilities 1-2 orders of magnitude lower than commercial RO membranes and salt permeabilities two orders of magnitude higher. It is unclear how the insights of this work will contribute to the development of improved membranes since performance of even the optimal membrane is orders of magnitude behind conventional polymers. Although the results give helpful insights into this specific system, it is unclear how they would apply to other nanofluidic processes. Membrane compaction effects are already well-documented in polymeric membranes and the studied effects seem somewhat comparable.

Response:

Thanks for the reviewer's comments. These concerns are indeed necessary to be addressed.

For the performance of GO-based membrane, theoretical simulation has revealed ultimate performance of two-dimensional-material membranes than current ones. In addition, theoretically the two-dimensional-material membranes can obtain the minimal membrane thickness (atomic thickness) for separation membranes to minimize transport resistance, which are the distinct advantages over other membrane materials. The studies on two-dimensional-material membranes just begun, and there are still a lot of fundamental challenges and technical issues to be addressed for structural optimization and mechanistic understandings. The number of publications reporting GO membranes for NaCl rejection is quite limited compared to the separation of other relatively larger solutes, which is also the motivation for us to conduct this study. This study focused on the mechanism of membrane nanochannel inconsistency during membrane process. The insights into the mechanism can help to optimize membrane structure and improve membrane performance, which contributes to the realization of ultimate performance predicted by theoretical works. Realizing this process may take a long time, which is also similar to the case of polyamide RO membranes. The current performance of polyamide membranes was achieved through intensive efforts over more than half century. We have included above discussion and provided a list of GO-based membrane RO performance for reference in the revised manuscript.

For the “*specific system*” and “*other nanofluidic process*”, the phenomenon found in GO nanochannel under RO could also provide some useful information of maintaining channel confinement for other two-dimensional-material laminates (e.g., Ti₃C₂T_x, boron nitride, carbon nitride) applied in other nanofluidic process. While the material and application investigated in this work are specific, we believe the information of what affects channel confinement and how to maintain it are also interesting for other systems. For other two-dimensional-materials, their membranes usually have problems in the structural stability due to weak interactions between nanosheets. For instance, laminated MXene (Ti₃C₂T_x) membranes also suffered from instability and chemical modification or crosslinking are usually adopted to address this issue for ion rejection (*Nat. Sustain.* 2020, 3, 296; *ACS Nano* 2019, 13, 9, 10535).

Similar requirements are also necessary for other pressurized membrane processes including organic solvent separation (*Nat. Commun.* 2018, 9, 1902) and gas separation (*Angew. Chem. Int. Ed.* 2019, 58, 13969). In addition to membrane process, the deformation constraint to maintain the channel confinement is important for other nanofluidic processes involving aqueous chemistry within nanometric slit pores (*Chem. Rev.* 2021, 121, 11, 6293). For instance, the nanochannel confinement enabled controlled ion transport revealed great potential for sustainable power generation (*Nat. Commun.* 2019, 10, 3876); the confinement of fluid in laminates boosted radical yields for efficient advanced oxidation processes (*Angew. Chem. Int. Ed.* 2019, 58, 8134). We have included above discussion to provide information on how the findings in this work could be applied in other nanofluidic process.

For the well-documented membrane compaction effects in polymeric membranes, though the observed swelling effect is similar between polymeric and GO membranes, the structural changes after swelling are much different. Two-dimensional-material laminates have well-defined nanochannels, which are quite different from the free volume generated by chain mobility in polymers. The factors affecting the structural stability of laminates thus could be distinct, as we found in this work the chemistry of both nanosheet and confined molecules between nanosheets had impacts. Revealing the distinct properties in laminates is critical to construct stable and tunable nanochannels for either studying molecule transport or nanofluidic applications. We have included this discussion in the revised manuscript.

The above discussions have been added in the context of the revised manuscript to present our results and discussion from a more general perspective:

Manuscript (RESULTS – Nanochannel confinement in water)

“...Both the occupation of Glc molecules in the channels and the reduced interlayer spacing hindered water transport (simulation results in **Supplementary Figure 12** and **Supplementary Movies 5 and 6**), resulting in a decreased water flux, particularly when including a possibility of more Glc molecules in the nanochannel.

Although the observed swelling and compaction phenomena of the GO nanochannel membrane are similar to those of the polymeric membranes, the structural changes after swelling are significantly different. 2D material laminates have well-defined nanochannels, which are quite different from the free volume generated by the chain mobility in polymers. The factors affecting the structural stability of laminates could thus be distinct, as we observed that the chemistry of both the nanosheet and confined molecules between nanosheets had an impact. Determining the properties of laminates is important for constructing stable and tunable nanochannels for studying the mass transport and performance improvement in nanofluidic applications.”

Manuscript (RESULTS – Role of the deformation constraint of nanochannel for salt/water separation)

*“...Long-term performance evaluation (>150 d) at 3 MPa was conducted using various NaCl feed concentrations ranging from 500 to 5000 ppm (**Figure 6c**). Each feed concentration was tested for approximately one month, and the membrane exhibited stable NaCl rejection throughout the entire *period*. The membrane flux gradually decreased because of the increased osmotic pressure at high NaCl concentrations *and* possible membrane compaction. However, the optimal membrane, possessing a strong constraint on nanochannel deformation, demonstrated promising results. While this study aims to investigate the nanochannel deformation using Glc, possessing no distinct interactions with the GO, the obtained stable NaCl rejection performance is still comparable to studies on GO-based membranes (**Supplementary Table 3**). The findings of this study could aid in designing modifications and intercalations for GO membranes to achieve nanochannel deformation constraints and satisfactory flux and salt rejection. For example, grafting agents that possess distinct chemical structures and functional groups (such as π -conjugated molecules with water-affinity functional groups) can stabilize the nanochannel and favor water diffusion, which should work better than the weakly interacted Glc demonstrated in this study.”*

Supplementary Information

“**Supplementary Table 3.** Summary of literature reports on GO-based membranes exhibiting relatively high NaCl rejection.”

Membrane code	Permeance [L m ⁻² h ⁻¹ bar ⁻¹]	Feed NaCl concentration [ppm]	NaCl rejection [%]	Reference
PE@ArGO	2.9	58.4	88	2
GO-FLG	0.35	2000	85	3
GO (EPR)	25	500	97	4
GNM/SWNT	110.6	2000	85	5
TU-GO	1.5	100	95	6
TCPP-rGO	0.17	500–2000	85–92	7
Air-rGO	1.05	1000	83	8
K-rGO	0.6	1000	91	9
GO-TBO	0.4	584–29220	30–85	10
TA-rGO	0.36	500–2000	80–92	11
GO/g-C ₃ N ₄	33.5	1170	87	12
GO-AQP	7.83	1000	99.1	13
Glc-rGO (This study)	0.14	500–5000	88.7–93.2	

Manuscript (DISCUSSION)

“In summary, we investigated *the* nanochannel deformation behavior of multilayered GO membranes and determined *the* balance between the channel chemistry and confinement stability for the reverse osmosis of salt solutions. The deformation *depended* on the pore wall (GO nanosheets) and pore interior (intercalated moieties) of the nanochannel. Notably, while minor changes to the pore interior *negligibly altered the* channel dimensions from *typically* characterized values, they may result in a significant difference in the nanochannel stability (deformation constraint) during reverse osmosis. *Thus, the discord* between the characterization and performance of *the* GO nanochannel membranes *was* highlighted. *The effect of the intercalation chemistry should be carefully considered when designing nanochannel membranes using functionalized nanosheets to construct* of confined nanochannels. *There is still a*

considerable performance gap between the state-of-the-art GO-based membranes and commercial polyamide membranes for reverse osmosis. Therefore, future efforts are required to improve the quality of raw GO materials and search for effective modification agents. Suitable modification agents that can constrain the swelling or compaction of laminates and exhibit low transport resistance in nanochannels are desirable for maintaining constrained nanochannels for precise ion separation. In addition, large-scale production of high-quality GO nanosheets is expected in future to make the fabrication of few-layered laminates more practical for maximizing water flux.

The phenomenon observed in the GO membrane nanochannels under reverse osmosis could also provide useful information for maintaining channel confinement for materials of other laminates and applications of other nanofluidic processes. Although the material and application investigated in this study are specific, we believe that the results on what affects channel confinement and how to maintain it are also applicable to other systems. For example, the nanochannel deformation and its constraint could be extended to laminate membranes, including MXene (Ti_3C_2Tx)^{37,38}, boron nitride^{39,40}, and carbon nitride⁴¹, where functionalization is necessary to achieve stable nanochannels during pressurized membrane separation processes in water, organic solvents, or gas. In addition to the membrane process, the factors affecting the channel confinement constraint are also relevant for other nanofluidic processes involving aqueous chemistry within nanometric slit pores⁶. For example, nanochannel confinement enables controllable ion transport in laminates, exhibiting significant potential for salinity gradient power generation⁴² in the water-energy nexus, and fluid confinement in laminates boosted radical yields for efficient chemical reactions⁴³ in advanced oxidation processes. However, it is recommended that the consistency of channel confinement in nanofluidic applications be evaluated by designing experiments based on the characteristics of specific processes, thereby acquiring incremental insights into the nanochannel deformation behavior of different materials and applications.”

- **Comment 2:**

The mechanisms shown in Figure 1 and 3 (also described in line 143) where water influx occurs and water squeezes out seem speculative and it is unclear how they are supported by the data. From the explanation of the authors, it seemed like the membranes are perturbed at high pressures (and become more permeable and less selective) before stabilizing.

Response:

Thanks for the reviewer's comment. The illustrated deformation processes in Figure 1 and Figure 3 were given according to the computational simulation results and time-course water flux situations in experiments. The illustrations in Figure 1 and Figure 3 were shown as the schematics of perturbed laminate structure to present our research ideas and perturbed nanochannels to indicate the results from simulation, respectively. We have modified the illustration in Figure 1 to only show the concept of possible perturbed structure of laminates under pressure. The illustration in Figure 3 and the description on water influx and squeezing out in context were removed. Revised Figure 3 now only shows the snapshots of laminate structure evolution from simulation results to reflect the channel deformation process. Additional simulation was also performed to identify the interactions between GO and Glc molecules. We have modified relevant part in the revised manuscript:

Manuscript (INTRODUCTION)

*"...Typically, pressurized processes, such as reverse osmosis (see illustration in **Figure 1**), require high pressures, necessitating the robust tolerance²⁸ of the nanochannels formed in the laminates (**Figure 1**) to minimize perturbed structure-rendered low salt rejection. Hence, it is vital to understand the response and dynamic behavior of nanochannels while transporting pressure-driven flows is essential."*

“Figure 1. *Illustrations of reverse osmosis system, multilayered nanosheet membrane, and interlayered nanochannels.* During reverse osmosis, external pressure is applied to force water molecules through a semipermeable membrane. Nanochannels are then formed by neighboring nanosheets in the multilayered nanosheet membrane. Different perturbation of nanochannels with different deformation constraint occurs in the pressurized flow of water and salt.”

Manuscript (RESULTS – Nanochannel confinement in water)

“...The resulting expansion of the interlayered *nanochannels* may prevent effective ion sieving. Additionally, when the wet nanochannel was subjected to a pressurized water flow, the channel *could be perturbed with increased permeance but decreased salt rejection.*”

Molecular dynamics (MD) simulations were performed to investigate the evolution of the laminate structure under water pressure conditions and to reveal the contribution of Glc to the structural stability (Figure 3a, Supplementary Figure 9, and Supplementary Movies 1–4).”

“Figure 3. **Nanochannel deformation in water.** (a) MD simulations of laminate models of GO, rGO, and Glc-rGO in pressurized water. Pressures of 50 and 10 MPa were applied on the opposite sides of the simulation cell for all the models, as indicated in the GO model as an example. Two Glc-rGO models with two and six Glc molecules in each channel were constructed, denoted as Glc(2)-rGO and Glc(6)-rGO, respectively; (b) $\text{sign}(\lambda_2)\rho$ mapped 2D scatter plot of the intermolecular electron density gradient difference (δg_{inter}). The value of $\text{sign}(\lambda_2)\rho$ is represented by filling color according to the color bar. ρ denotes the electron density at the weak interaction critical point, $\text{sign}(\lambda_2)$ denotes the sign of the second largest eigenvalue λ_2 of the electron density Hessian matrix; (c) Isosurface of δg_{inter} with an isovalue of 0.005 a.u.. The blue isosurface between the oxygen-containing group of rGO and the hydroxyl group of Glc indicates the C-H...O hydrogen bond interactions as depicted by dashed lines; (d) XRD spectra of wetted GO and Glc-rGO samples (dashed lines and arrows indicate peak shifts); (e) Time-course pure water flux of GO and Glc-rGO samples at 1 MPa. Glc ratio for Glc-rGO samples derived from different reaction periods (T0–T80) was fixed at 10 (Glc10), whereas the reaction period for Glc-rGO samples derived from different Glc ratios (Glc0–Glc20) was fixed at 60 min (T60).”

- **Comment 3:**

It is unclear how the properties of the Glc10T20 membrane were improved after deformation (line 291). The data appears to show that rejection and molecular weight cutoff remained constant.

Response:

We are sorry that we did not make it clearly presented. In this experiment, we observed that the organic solute rejection and MWCO value of Glc10T20 was improved. For example, Glc10T20 showed ~20% and ~40% rejection of the neutral solute glycerol (92 Da) before and after pressurization, respectively. The MWCO also became smaller from ~350 Da to ~200 Da after pressurization. We have modified the plot to present the data more clearly. In addition, we have fitted the pore radius distribution curve of Glc10T20 expressed by probability density function using the data from MWCO curve. From the pore radius distribution results, the difference before and after pressurization could be more obvious to be distinguished.

We have included above results and discussion in the revised manuscript and Supplementary Information:

Manuscript (RESULTS – Role of the deformation constraint of nanochannel for salt/water separation)

*“The molecular weight cut-off (MWCO, defined as the lowest molecular weight solute in which **the membrane retains 90% of the solute**) of these membranes was **investigated** to reveal their separation capability **toward neutral solutes and channel dimensions** before and after the pressure **variations**. Rejection tests for **organic solutes with a suitable molecular weight** ³⁶, including glycerol (92 Da), Glc (180 Da), sucrose (342 Da), and raffinose (504 Da), were conducted to obtain MWCO curves (**Figure 5g**). Before **the** pressure variation, the MWCO values of the different membranes were ~180–500 Da, and the order corresponded to **the** NaCl rejection results. After pressure*

variations, nanochannel deformation occurred as the MWCO values of GO, Glc10T20, and Glc20T60 changed, whereas the optimal Glc10T60 remained stable (Supplementary Table 2). GO and Glc20T60 were transformed into less confined nanochannels, whereas Glc10T20 improved. The pore radius distributions of GO and Glc10T20 (Supplementary Figure 18) were attainable according to their MWCO curves, where the GO exhibited similar results after the pressurization, whereas Glc10T20 exhibited a narrower distribution and smaller pore radius. This indicates that pressurization can either improve or deteriorate the deformable nanochannel properties for separating molecules within a certain size range. From the results...”

“Figure 5. Nanochannel deformation and performance in pressure-changing reverse osmosis. Filtration of NaCl solutions with pressure variations by (a) GO, (b) Glc10T20, (c) Glc10T60, and (d) Glc20T60; (e) Glc content in original membrane precursor suspension, intercalated Glc content in prepared membranes, and leached Glc content from membranes during permeation test. The error bars represent the standard

deviations of three measurements; (f) Illustration of Glc leaching from laminates with the pressurized flow; (g) Molecular weight cut-off curves of membranes before and after pressure variations. The error bars represent the standard deviations of three measurements.”

Supplementary Information

“**Supplementary Table 2.** MWCO of different membranes before and after pressurization.”

	GO	Glc10T20	Glc10T60	Glc20T60
MWCO before	480 Da	311 Da	180 Da	352 Da
MWCO after	>500 Da	188 Da	180 Da	>500 Da

“**Supplementary Fig. 18.** Fitted pore radius distribution of GO and Glc10T20 membranes.”

“The pore radius distribution curve was expressed as a probability density function.¹ The mean pore radius of the membrane was assumed to be the Stokes radius of the organic solute with a measured rejection of 50%. The geometric standard deviation of the distribution is assumed to be the geometric standard deviation of the probability density function curve, which is the ratio of the Stokes radius with a rejection of 84.13%

to that with a rejection of 50%. The pore radius distribution of the membrane can thus be expressed as:

$$\frac{dR(r_p)}{dr_p} = \frac{1}{r_p \ln \sigma_p \sqrt{2\pi}} \exp \left[-\frac{(\ln r_p - \ln \mu_p)^2}{2(\ln \sigma_p)^2} \right]$$

where r_p denotes the Stokes radius of the organic solute, σ_p the geometric standard deviation of the distribution curve, and μ_p the mean pore radius.

The Stokes radii of organic solutes can be calculated as

$$\ln(r_p) = -1.4962 + 0.4654 \ln(M_w)$$

where M_w denotes the molecular weight of the organic solute.”

• **Comment 4:**

The paper frequently refers to the membrane system as nanochannels. I am not aware of the conventions in this field, but the system is appears to be an extremely heterogenous network of angstrom-scale voids between the sheets.

Response:

The nanochannel is referred to the nanochannel in membranes, which is the void between stacked nanosheets for laminates. The nanochannel is mainly referred in this paper to make an emphasis on the nanoscale structure and chemistry of this kind of void for easier description of the nanosheet displacement-rendered changes in voids between nanosheets. Such use of ‘nanochannel’ for membrane system has also been used in other papers: *Chem. Soc. Rev.*, 2018, 47, 322; *Chem. Soc. Rev.*, 2020, 49, 1071; *Nat. Commun.*, 2019, 10, 5793; *Nat. Commun.*, 2021, 12, 507. According to the reviewer’s comment, we have added clarification and revised the corresponding expression of ‘nanochannel’ as “membrane nanochannel” at relevant places where the membrane system is referred. Below are some examples and all the modifications can be found in the revised manuscript:

Manuscript (ABSTRACT)

“...We built a series of *membrane nanochannels* with similar physical dimensions but different...”

Manuscript (INTRODUCTION)

“Nanochannel-confined transport can be easily performed in the interlayers of laminates *using stacked two-dimensional (2D) materials*, such as graphene oxide (GO) nanosheets. Multilayered GO nanosheets *form a heterogenous network of angstrom-scale voids between the sheets, which function as membrane nanochannels for transport and separation*. This nanochannel separation membrane can be fabricated by a *practically feasible* way...”

“Additionally, while pressure-driven filtration is widely used for the separation evaluation of GO *membrane nanochannels*, only the...”

“To investigate the role of inconsistent nanoconfinement in nanochannel membranes during typical pressure-driven desalination, we prepared *GO-membrane nanochannels* with...”

• **Comment 5:**

Figure 1a is difficult to interpret. The left schematic show osmotic pressure and applied pressure simultaneously but it is unclear whether the authors are presenting the schematic at the equilibrium conditions (which is when this would apply). The meaning of the triangle in the nano channel is also unclear. Figure 2a may also be difficult for the reader to interpret. I would have assumed that the more reduced sample should have less oxygenated groups.

Response:

Thanks for the comments on the schematics and illustrations.

For the schematic of reverse osmosis process in **Figure 1**, it aims to provide general information about reverse osmosis process which illustrates the applied pressure to overcome osmotic pressure of salt solution to obtain pure water at the permeate side.

We have simplified the denotes and removed the illustration of equilibrium condition in the schematic to avoid misleading.

For the meaning of triangle, it indicates the salt solutes in water, as we put a legend below the reverse osmosis illustration. We have changed the symbol representing salt solutes to be more recognizable green circles and enlarged the legend to make this clearer.

For the **Figure 2a**, it aims to indicate that the interlayer spacing is dependent on both reduction degree of GO and intercalated amount of Glc. We have modified the illustration for better interpretation (that is, reducing GO leads to smaller interlayer spacing while intercalating Glc leads to larger one).

The revisions are as follows:

Manuscript (INTRODUCTION)

“Figure 1. Illustrations of reverse osmosis system, multilayered nanosheet membrane, and interlayered nanochannels. During reverse osmosis, external pressure is applied to force water molecules through a semipermeable membrane. Nanochannels are then formed by neighboring nanosheets in the multilayered nanosheet membrane. Different perturbation of nanochannels with different deformation constraint occurs in the pressurized flow of water and salt.”

Manuscript (RESULTS – Controlling GO membrane nanochannels)

“In this study, Glc, which does not exhibit solid interplay with GO, was used as the reducing agent to observe the trade-off between GO reduction and Glc intercalation. GO reduction removes its oxygenated functional groups and narrows the dimensions

(improves the confinement) of the interlayered nanochannel between neighboring nanosheets, whereas the possibly intercalated Glc in the laminates induces an opposite effect on the channel dimensions (Figure 2a). Chemical reactions between GO and Glc...”

“Figure 2. **Fabrication of different GO membrane nanochannels.** (a) Illustration of varied GO membrane nanochannel dimensions by reaction with Glc; (b) Chemical reaction of GO (hydroxyl) reduction by Glc; (c) XPS spectra of GO and a typical Glc-rGO sample (reaction period, 60 min; Glc ratio, 10); (d) O/C ratio of GO and different Glc-rGO samples from XPS analysis. The error bars represent the standard deviations of three measurements; (e) X-ray diffraction spectra of GO and Glc-rGO samples (dashed lines and arrows indicate peak shifts). The Glc ratio for the Glc-rGO samples derived from different reaction periods (T0–T80) was fixed at 10 (Glc10), whereas the reaction period for the Glc-rGO samples derived from different Glc ratios (Glc0–Glc20) was fixed at 60 min (T60).”

• **Comment 6:**

It is unclear how the slip flow described in the introduction is relevant for the system described.

Response:

Thanks for the comment. We gave the example of slip flow in Introduction to highlight the distinct properties found for confined channels. This work is not targeted at the study of slip flow phenomenon. Sorry for the misleading and we have revised the context into a more general statement in the Introduction section:

Manuscript (INTRODUCTION)

“...e.g., metal-organic frameworks, carbon nanotubes, and nanosheets provide zero-, one-, and two-dimensional cavities or channels at the sub-nanometer and nanometer scales. Fluid transport confined in nanometric dimensions has drawn significant attention ^{6, 7, 8, 9, 10, 11} as it displays distinct behaviors that are considerably different from those in a bulk system. Multiple factors, including the chemical nature and dimensionality of the confining nanopore interacting with the altered properties of the confined fluid, may lead to unexpected liquid transport ¹². This is particularly relevant for improved separation at the molecular scale, exhibiting significant potential for rapid transport and precise differentiation.”

Comments from Referee 3

General Comment:

This paper is very important for understanding the microstructure evolution and filtration efficiency of assembled membrane of two-dimensional materials under pressure, and the authors need to clarify the following issues before publication.

Response:

Thanks for the reviewer's evaluation. We appreciate your recognition of this research work. The revisions have been included in the revised manuscript according to the reviewer's valuable comments. Please find below a point-by-point response to the comments and concerns.

- **Comment 1:**

In the figure3(b), the diagram should show how the pressure is applied. As shown in the figure S8, the structural are not force-balanced, how is the GO film supported without overall moving under the action of pressure? Boundary constraints for GO in MD should be clarified in the schematics and methods part.

Response:

Thanks for the reviewer's question and comments.

For the diagram, the Figure 3(b) has been modified to show how the pressure is applied. Denotes have also been added in the title of the figure.

For the force-balance, the GO nanosheets actually have displacement in the direction parallel with the external force. The purpose of applying the unbalanced force (i.e., the non-equilibrium MD simulation) in the current study is to mimic the condition that the membrane is squeezed under the pressure, and then study the behavior of the membrane during this process. The similar non-equilibrium MD simulation by applying a higher pressure at one side than the other has been successfully used to deal with

various membrane related topics (*ACS Appl. Mater. Interfaces*, 2017, 9, 22826; *J. Phys. Chem. C*, 2016, 120, 22585; *J. Phys. Chem. Lett.*, 2010, 1, 1590; *J. Membr. Sci.*, 2015, 496, 108). We have included this information in the revised manuscript.

For the boundary constraints for the laminate stability simulation, same boundary constraints were used as in the simulation for water transport in laminates. We have clarified it in the METHODS section of the manuscript.

The revisions in the manuscript are as follows:

Manuscript (RESULTS – Nanochannel confinement in water)

“Figure 3. **Nanochannel deformation in water.** (a) MD simulations of laminate models of GO, rGO, and Glc-rGO in pressurized water. Pressures of 50 and 10 MPa were applied on the opposite sides of the simulation cell for all the models, as indicated in the GO model as an example. Two Glc-rGO models with two and six Glc molecules in

each channel were constructed, denoted as Glc(2)-rGO and Glc(6)-rGO, respectively; (b) $\text{sign}(\lambda_2)\rho$ mapped 2D scatter plot of the intermolecular electron density gradient difference (δg_{inter}). The value of $\text{sign}(\lambda_2)\rho$ is represented by filling color according to the color bar. ρ denotes the electron density at the weak interaction critical point, $\text{sign}(\lambda_2)$ denotes the sign of the second largest eigenvalue λ_2 of the electron density Hessian matrix; (c) Isosurface of δg_{inter} with an isovalue of 0.005 a.u.. The blue isosurface between the oxygen-containing group of rGO and the hydroxyl group of Glc indicates the C-H...O hydrogen bond interactions as depicted by dashed lines; (d) XRD spectra of wetted GO and Glc-rGO samples (dashed lines and arrows indicate peak shifts); (e) Time-course pure water flux of GO and Glc-rGO samples at 1 MPa. Glc ratio for Glc-rGO samples derived from different reaction periods (T0–T80) was fixed at 10 (Glc10), whereas the reaction period for Glc-rGO samples derived from different Glc ratios (Glc0–Glc20) was fixed at 60 min (T60).”

Manuscript (METHODS – Computational simulation details)

“To investigate the stability of the laminate structure in water under applied pressure, four models were constructed including GO, rGO, Glc(2)-rGO (two glucose molecules inserted into each channel), and Glc(6)-rGO (six glucose molecules inserted into each channel) (**Supplementary Figure 9**). The GO and rGO sheets were built based on the Lerf-Klinowski model ⁴⁵, in which the hydroxyl and epoxy groups were randomly distributed on both sides of the carbon basal plane. The O/C ratio and interlayer spacing of GO/rGO were determined by matching the values obtained from practical samples. For the GO sheet, the O/C ratio was ~ 0.6 , and the interlayer spacing was 8.03 Å. For the rGO sheet, the O/C ratio was ~ 0.3 , and the interlayer spacing was 7.62 Å. The membrane was composed of trilaminar GO/rGO sheets with a slit of ~ 6 Å in the x–y plane. The offset between the slits of the GO/rGO sheets was 29.5 Å. Different numbers of Glc molecules intercalated in the constructed laminate models were employed to explore the effect of the Glc. Two rigid graphene sheets were placed at the ends of the cell. The space between the two graphene sheets was filled with water. A vacuum space of ~ 40 Å along the z-direction was applied to prevent interactions between the system and its images. Pressures of 50 and 10 MPa were applied to the two graphene sheets. The size of the orthogonal cell of the four models was $61.5 \times 29.8 \times 150.0$ Å³ (x, y, z) (**Supplementary Figure 20a**). In the current study, the purpose of

applying the unbalanced force (that is, the nonequilibrium MD simulation) is to mimic the condition in which the membrane is squeezed under the pressure and then investigate the behavior of the membrane during this process. Similar nonequilibrium MD simulations have been successfully employed by applying a higher pressure on one side than on the other to deal with various membrane related topics ^{46, 47, 48, 49}.

To investigate the water transport through the laminates of GO and Glc-rGO, two models were constructed: GO and Glc(6)-rGO (six Glc molecules inserted into each channel) (Supplementary Figure 12). In the two models, an orthogonal system cell of $61.5 \times 29.8 \times 119.1 \text{ \AA}^3$ (x, y, z) was used with periodic boundary conditions applied in all directions. The membrane comprised trilaminar GO/rGO sheets with a slit of $\sim 6 \text{ \AA}$ in the x-y plane. The offset between the slits of the GO/rGO sheets was 29.5 \AA . One rigid graphene sheet was placed 40 \AA from the nearest GO/rGO sheet. The chamber between the graphene sheet and the nearest GO/rGO sheet was filled with water. The vacuum chamber on the other side of the membrane was 40 \AA along the z-direction (Supplementary Figure 20b). A pressure of 150 MPa was applied to the graphene sheet to facilitate the water transport through the laminates.

All MD simulations were performed...”

Supplementary Information

“Supplementary Fig. 20. Boundary constraints applied for studying the (a) laminate structure stability, and (b) water transport.”

- **Comment 2:**

In the line129-131, the authors suggest that Glc intercalation effect nanosheet stacking. Could the authors explain with more clear description how the intercalated Glc affect the evolution of the peak shift with the reaction time.

Response:

Thanks for the comment. While same amount of Glc molecules was used for T“y” membrane, different reaction time induced different reduction degree of GO, which might affect the interactions between Glc and nanosheets and nanosheet stacking order. Initially (from T0 to T40), the peak shifted to higher 2-theta values (smaller interlayer spacing) because of the loss of oxygenated groups of GO; Afterwards (from T60), Higher reduction degree of GO might induce worse compatibility with Glc molecules causing unfavorable stacking during filtration-induced assembly and enlarging the overall interlayer spacing. The effect of Glc intercalation to expand the interlayer was excluded for T“y” laminates by analyzing the finally intercalated Glc amount in the laminates.

We have provided more discussion on the effect of reaction time on the evolution of the peak shift in XRD spectra. We also used laminate samples subjected to XRD analysis with higher nanosheet loading amount to double-confirm the peak position results. Below are the revisions:

Manuscript (RESULTS – Controlling GO membrane nanochannels)

“With an increasing Glc ratio, the peak position did not significantly change from Glc1 to Glc15 (the peak intensity of Glc20 was too low to assign). Clearer characteristic peaks could be observed in laminate samples with increased deposition amount, and similar peak shifting tendencies with a change in the reaction conditions

were confirmed (**Supplementary Figure 8**). These results tentatively reveal the Glc intercalation effect during nanosheet stacking, as a higher reduction degree is typically accompanied by a continuous peak shift to higher 2θ values¹⁹. Although the same amount of Glc molecules was used for T'y' membrane, different reaction periods induced different reduction degree of GO, which affected the interactions between the Glc and the nanosheets and nanosheet stacking order. A higher reduction degree of GO may induce worse compatibility with the Glc molecules, causing unfavorable stacking during filtration-induced assembly and increasing the overall interlayer spacing. A higher amount of Glc used would lead to a higher reduction of GO in the Glc'x' membranes, decreasing the interlayer spacing. However, higher amount of Glc may also have a higher possibility of retaining more Glc molecules, increasing the interlayer spacing. Consequently, this trade-off may restrict the interlayer spacing change, maintaining a similar peak position, as shown in the XRD spectra. The shift in the peak position was limited to the narrow range of 11–12° (**Supplementary Table 1**), implying that the dimension change of the nanochannel after the reaction was insignificant. Therefore, *investigating* the impact of minor changes in the nanochannel on channel properties under reverse osmosis conditions *would be beneficial*.”

“**Supplementary Fig. 8**. XRD spectra of thicker GO and Glc-rGO samples (dashed lines and arrows indicate peak shifts). These thicker samples were prepared by using 5-fold GO or Glc-rGO deposition amounts of normal samples.”

Manuscript (RESULTS – Role of the deformation constraint of nanochannel for salt/water separation)

*“The improved hydrophobicity of these three Glc-rGO membranes (**Supplementary Figure 11**) indicates the improved water stability of their bulk structures. This was inconsistent with their nanochannel stability and salt rejection, as they had different responses to the infiltrated water flow in their different pore interiors. The amount of Glc retained after the preparation of these laminates could be different, considering the different degrees of GO reduction and interactions with Glc. The amount of Glc intercalated was similar for the Glc10T20 and Glc10T60 laminates, whereas it was higher for Glc20T60 (**Figure 5e** and **Supplementary Figure 17**), indicating that higher amounts of Glc used in suspension lead to a higher possibility of retained Glc in the corresponding laminates. Although this explains the difference in performance between Glc10 and Glc20, the difference between T20 and T60 must be further investigated.*

*As Glc molecules have hydrophilic properties, the intercalated Glc molecules can also be dissolved with pressurized flow during the permeation test (**Figure 5f**). The leached amounts of Glc for these membranes during permeation tests were in the order of Glc10T20 < Glc10T60 < Glc20T60 (5%, 15%, and 24%, respectively; **Figure 5e**). This order is also consistent with their flux decline tendency, indicating that the presence of Glc molecules in the nanochannels was influenced by channel stability. Higher Glc leaching indicates lower integration of Glc and rGO and, thus, a higher possibility of a perturbed structure during Glc leaching by pressurized flow. The mutable and dynamic GO nanochannels cannot function like rigid pores in response to external pressure and NaCl in the feed, to strictly exclude hydrated ions from water. Nanochannel deformation frequently occurs when external conditions are changed. An improved deformation constraint can be obtained by optimizing the chemistry of the GO nanosheets and avoiding the excessive intercalation of undesirable molecules into multilayered nanochannels to maximize the stable rejection of ions.”*

“Figure 5. **Nanochannel deformation and performance in pressure-changing reverse osmosis.** Filtration of NaCl solutions with pressure variations by (a) GO, (b) Glc10T20, (c) Glc10T60, and (d) Glc20T60; (e) Glc content in original membrane precursor suspension, intercalated Glc content in prepared membranes, and leached Glc content from membranes during permeation test. The error bars represent the standard deviations of three measurements; (f) Illustration of Glc leaching from laminates with the pressurized flow; (g) Molecular weight cut-off curves of membranes before and after pressure variations. The error bars represent the standard deviations of three measurements.”

• **Comment 3:**

In the figure2(e) and figure3(c), for a better understanding, both the angle and corresponding interlayer distance could be present.

Response:

Thanks for the comment. We have summarized the peak positions and corresponding d-spacing values for XRD analysis and included a table in the revised Supplementary Information:

“Supplementary Table 1. Summary of characteristic peak positions and corresponding d-spacings from the XRD analysis. The data for the Glc10T80 and Glc20T60 samples are not provided because their peak intensities were too low to be assigned.”

Laminates	Peak position (2θ , °)			d-spacing (Å)		
	dry	wet	wet (NaCl)	dry	wet	wet (NaCl)
GO	11.0	7.0	7.2	8.04	12.62	12.27
T'y'	Glc10T0	11.3	7.2	7.2	7.82	12.27
	Glc10T10	11.5	7.2	7.8	7.69	12.27
	Glc10T20	11.6	11.5	8.5	7.62	7.69
	Glc10T40	11.8	11.6	10.0	7.49	7.62
	Glc10T60	11.7	11.3	10.7	7.56	7.82
	Glc10T80	–	–	–	–	–
Glc'x'	Glc0T60	11.7	11.4	8.5	7.56	7.76
	Glc1T60	11.4	11.4	9.0	7.76	7.76
	Glc5T60	12.0	11.7	10.1	7.37	7.56
	Glc10T60	11.7	11.3	10.7	7.56	7.82
	Glc15T60	11.6	11.4	11.0	7.62	7.76
	Glc20T60	–	–	–	–	–

• **Comment 4:**

As the description in the lines 252-257(pages 14-15), in the initial pressurization process, the permeation flux first increases and then decreases, finally the permeation flux tends to be stable in the subsequent pressure cycle. (1) Is there any irreversible deformation of the GO channel occurring in this process, and what exactly is the deformation. (2) As the number of pressure cycles increases, will the permeation flux and selectivity of the GO remain stable? Will the deformation change periodically under pressure cycles?

Response:

Thanks for the questions.

For the question (1), there is irreversible deformation occurred in this process for relatively unstable samples of GO, Glc10T20, and Glc20T60 but not for the stable one of Glc10T60. The changes can be evidenced from NaCl solution filtration data and MWCO test. For the NaCl rejection test, we applied pressure with the sequence of 1 MPa → 2 MPa → 1 MPa → 2 MPa → 1 MPa. By comparing the 3 cycles of permeation results at 1 MPa, we could see there were some differences of NaCl rejection for some laminates; For the MWCO curves, except Glc10T60, other laminates showed different MWCO values after pressurization process which indicates the changed channel selectivity. The deformation could be the displacement of nanosheets in the laminates due to repeated swelling and compaction during pressurization and depressurization process. We have provided supplementary figures and more discussions to clarify this.

For the question (2), deformation change will not be periodical, and filtration performance will become more and more stable after more testing cycles. As found in the NaCl rejection testing cycles (1 MPa → 2 MPa → 1 MPa → 2 MPa → 1 MPa), for relatively unstable laminates, the permeation flux sharply increased and then slowly decrease when the applied pressure initially increased from 1 MPa to 2 MPa during the first cycle. However, during the second cycle of 1 MPa → 2 MPa, the permeation flux tended to become more linearly dependent on the increased pressure, indicating a more stable structure. The rejection performance also showed similar trend. Therefore, applying more testing cycles will not give periodic change of performance as that found in the first cycle. We have included above discussion in the revised manuscript.

Below are the revisions made in the revised manuscript and Supplementary Information:

Manuscript (RESULTS – Role of the deformation constraint of nanochannel for salt/water separation)

“The Glc10T60 membrane with optimal nanochannel stability achieved the highest and most stable NaCl rejection (~90%) during filtration with pressure variation,

although its flux decline at increased pressures was higher than that of Glc10T20 (Figure 5c). However, the Glc20T60 membrane performed poorly with a NaCl rejection similar to that of Glc10T20 (~60%) but with a significantly lower permeation flux (Figure 5d), although these laminates remained intact after pressurized filtration (Supplementary Figure 15). The filtration performances of different membranes at the 1 MPa steps in the filtration cycle (three 1 MPa steps in the testing cycle of 1 MPa → 2 MPa → 1 MPa → 2 MPa → 1 MPa) were compared (Supplementary Figure 16) to investigate irreversible changes in the membrane properties. If the membrane structure is sufficiently stable, the observed flux and salt rejection performance should be similar under the same external conditions. However, NaCl rejections for the GO, Glc10T20, and Glc20T60 samples were compromised after the pressurization process, implying that certain structural deformations were applied during the pressurization and depressurization. The deformation could be the displacement of nanosheets in their laminates by swelling and compaction processes, altering the nanochannel confinement.”

Supplementary Information

“Supplementary Fig. 16. Average permeation flux and NaCl rejection of different membranes at three 1 MPa steps during pressure-changing reverse osmosis. The first

filtration at 1 MPa was the initial test, whereas the second and third filtrations were tested after the filtration at 2 MPa.”

“The flux of these membranes at different steps of 1 MPa was similar, whereas the NaCl rejection was compromised after the first pressurization process for the GO, Glc10T20, and Glc20T60 membranes. The Glc10T60 membrane exhibited optimal stability and similar NaCl rejection. After one cycle of pressurization (1 → 2 MPa), all membranes became more stable in rejecting NaCl in the second and third 1 MPa steps, exhibiting constant rejection values.”

- **Comment 5:**

In the lines 291-301(page 17), the results indicate that pressurization can both improve and deteriorate deformable nanochannel properties. For the Glc10T20 the separation performance is improved, while for the GO- and Glc20T60 membrane the performance is disrupted. The authors explain this may be resulted by the defects and disordered manner. Is there any randomness in the behavior of different membranes? Why do some of the points in the Figure5(e-f) have error bars and some don't?

Response:

Thanks for the questions. The MWCO curves obtained from different batches of samples were consistent, not occasional. Some of error bars were small and were hidden behind the corresponding symbols in the plot. Sorry for the misleading and we have revised the plot to show the error bars in the revised manuscript for this as well as other plots:

“Figure 5. **Nanochannel deformation and performance in pressure-changing reverse osmosis.** Filtration of NaCl solutions with pressure variations by (a) GO, (b) Glc10T20, (c) Glc10T60, and (d) Glc20T60; (e) Glc content in original membrane precursor suspension, intercalated Glc content in prepared membranes, and leached Glc content from membranes during permeation test. The error bars represent the standard deviations of three measurements; (f) Illustration of Glc leaching from laminates with the pressurized flow; (g) Molecular weight cut-off curves of membranes before and after pressure variations. The error bars represent the standard deviations of three measurements.”

Comments from Referee 4

General Comment:

Multilayered graphene oxide membranes have been intensively studied in recent years but are still faced with big challenges in reverse osmosis. This manuscript provided new insights and perspectives of confinement states of graphene oxide interlayer nanochannels and demonstrated good results in effectively constraining the nanochannel robustness. The inconsistency of nanochannel confinement was evidenced by both experiments and simulations and valid desalination performance was obtained. Overall, the manuscript is well organized and could be publishable after minor revision. Below please find some detailed comments:

Response:

Thanks for the reviewer's evaluation. We appreciate your recognition of this research work. The revisions have been included in the revised manuscript according to the reviewer's valuable comments. Please find below a point-by-point response to the comments and concerns.

• Comment 1:

Regarding the construction of simulation models for different GO laminates, how did the authors decide the GO parameters (O/C ratio, interlayer spacing, etc.) and the number of glucose molecules in the nanochannel?

Response:

Thanks for the question.

The O/C ratio of GO or rGO sheet and the interlayer spacing between GO or rGO sheets in laminates were determined to roughly match with the characterized values of practical samples from XPS and XRD results. For other parameters that cannot be

obtained from experimental results, we used typical values similar as that in other literature reports.

For number of Glc molecules inserted in each channel, we did not specifically determine the value. We explored the effect of different number of Glc molecules on the laminate stability and found that when inserted Glc molecules increased from 2 to 6, the simulated structural stability was improved.

We have included above information in the revised manuscript:

Manuscript (METHODS – Computational simulation details)

“To investigate the stability of the laminate structure in water under applied pressure, four models were constructed including GO, rGO, Glc(2)-rGO (two glucose molecules inserted into each channel), and Glc(6)-rGO (six glucose molecules inserted into each channel) (Supplementary Figure 9). The GO and rGO sheets were built based on the Lerf-Klinowski model ⁴⁵, in which the hydroxyl and epoxy groups were randomly distributed on both sides of the carbon basal plane. The O/C ratio and interlayer spacing of GO/rGO were determined by matching the values obtained from practical samples. For the GO sheet, the O/C ratio was ~0.6, and the interlayer spacing was 8.03 Å. For the rGO sheet, the O/C ratio was ~0.3, and the interlayer spacing was 7.62 Å. The membrane was composed of trilaminar GO/rGO sheets with a slit of ~6 Å in the x–y plane. The offset between the slits of the GO/rGO sheets was 29.5 Å. Different numbers of Glc molecules intercalated in the constructed laminate models were employed to explore the effect of the Glc. Two rigid graphene sheets were placed at the ends of the cell. The space between the two graphene sheets was filled with water...”

• **Comment 2:**

For desalination tests in Fig. 4b, how did the authors obtain the equilibrated results of water flux and salt rejection performance?

Response:

Thanks for the reviewer's question. The performance results of different membranes were obtained when the observed water flux and NaCl rejection became stable. We have included detailed information in the revised manuscript:

Manuscript (METHODS – Desalination performance evaluation)

“The membranes were sealed in a homemade cross-flow module with an effective membrane area of ~7.1 cm² for the pure water permeation and salt rejection tests. The membranes were compacted by pure water filtration at 1 MPa for 16 h before further rejection performance evaluations. Salt solutions containing 500 ppm Na₂SO₄, NaCl, MgSO₄, and MgCl₂ were prepared. A transmembrane pressure of 1 MPa was applied during the cross-flow filtration tests. Membrane performance, including water flux and salt rejection values, was obtained when the observed values became constant over time. Typically, a duration of 8–16 h (depending on the membrane permeability) was applied for the filtration test of each feed solution. For the NaCl rejection test...”

• **Comment 3:**

In Fig. 5e and 5f, why selecting glycerol, glucose, sucrose, and raffinose as solutes for rejection tests?

Response:

Thanks for the reviewer's question. These organic solutes are typically selected for MWCO tests of desalination membranes. These solutes are neutral and with a reasonable range of molecular weights, which are suitable for obtaining the MWCO values for our membranes. We have included the reason and provided references in the context:

Manuscript (RESULTS – Role of the deformation constraint of nanochannel for salt/water separation)

“The molecular weight cut-off (MWCO, defined as the lowest molecular weight solute in which the membrane retains 90% of the solute) of these membranes was

investigated to reveal their separation capability toward neutral solutes and channel dimensions before and after the pressure variations. Rejection tests for organic solutes with a suitable molecular weight³⁶, including glycerol (92 Da), Glc (180 Da), sucrose (342 Da), and raffinose (504 Da), were conducted to obtain MWCO curves (Figure 5g)...

Manuscript (REFERENCES)

“36. Liang Y, et al. Polyamide nanofiltration membrane with highly uniform sub-nanometre pores for sub-1 Å precision separation. *Nat. Commun.* **11**, 2015 (2020).”

• **Comment 4:**

It is suggested to calculate and summarize the interlayer spacing values of interlayer nanochannels under different conditions in this paper for clear reference.

Response:

Thanks for the comment. We have summarized the peak positions and corresponding d-spacing values for XRD analysis and included a table in the revised

Supplementary Information:

“**Supplementary Table 1.** Summary of characteristic peak positions and corresponding d-spacings from the XRD analysis. The data for the Glc10T80 and Glc20T60 samples are not provided because their peak intensities were too low to be assigned.”

Laminates	Peak position (2θ, °)			d-spacing (Å)		
	dry	wet	wet (NaCl)	dry	wet	wet (NaCl)
GO	11.0	7.0	7.2	8.04	12.62	12.27
T'y'	Glc10T0	11.3	7.2	7.2	7.82	12.27
	Glc10T10	11.5	7.2	7.8	7.69	12.27
	Glc10T20	11.6	11.5	8.5	7.62	7.69
	Glc10T40	11.8	11.6	10.0	7.49	7.62
	Glc10T60	11.7	11.3	10.7	7.56	7.82
	Glc10T80	–	–	–	–	–

	Glc0T60	11.7	11.4	8.5	7.56	7.76	10.39
	Glc1T60	11.4	11.4	9.0	7.76	7.76	9.82
Glc'x'	Glc5T60	12.0	11.7	10.1	7.37	7.56	8.75
	Glc10T60	11.7	11.3	10.7	7.56	7.82	8.26
	Glc15T60	11.6	11.4	11.0	7.62	7.76	8.04
	Glc20T60	–	–	–	–	–	–

REVIEWER COMMENTS

Reviewer #1 (Remarks to the Author):

The authors reinforced their manuscript, and it is more suitable to the publication. However, one more concern is still not resolved.

1. For the response to comment 3, the authors fail to clearly explain the reason for the enhanced stability of Glc-rGO membrane, merely presenting the computation results. The authors claim that the nature of the interaction between Glc and rGO was identified; however, those kinds of interactions are certainly predictable without computation. For the experimental results, they still argue that there should be a proper concentration of Glc without convincing data; nevertheless, they said that the high Glc concentration induces a poor stability. In my opinion, the justification of such "proper concentration" is one of the major points of this manuscript, and the probable cause of the improved stability should be provided in experiments. At least, the interaction itself should be supported.

Reviewer #2 (Remarks to the Author):

The authors have provided detailed responses that have done a commendable job of addressing my comments and comments from the other reviews. I was particularly pleased to see changes in some of the schematics, which I think will make the manuscript easier to interpret for readers. I do not have additional comments.

Reviewer #4 (Remarks to the Author):

The revised manuscript has been greatly improved. The authors have clarified all of my concerns. The present manuscript is suitable for publication in this journal.

Responses to Reviewers' Comments

Manuscript ID: NCOMMS-22-31241A

Manuscript type: Article

Title: Deformation constraints of graphene oxide nanochannels under reverse osmosis

Corresponding author: Hideto Matsuyama; Wanqin Jin

We are grateful to the reviewers for their insightful comments on our paper, which are extremely helpful in improving the quality of the manuscript. We have been able to incorporate changes to reflect the suggestions provided by the reviewers. The changes within the manuscript are highlighted in blue.

Please find below a point-by-point response to the reviewers' comments and concerns.

Comments from Referee 1

- **General Comment:**

The authors reinforced their manuscript, and it is more suitable to the publication. However, one more concern is still not resolved.

Response:

We thank the reviewer for the recognition of our previously revised manuscript and the valuable comments, which are very helpful in improving the scientific rigor and quality of the research paper. We have revised manuscript according to the reviewer's further comments. The changes within the manuscript are highlighted in blue. Please find below the response to the concern.

- **Comment 1:**

For the response to comment 3, the authors fail to clearly explain the reason for the enhanced stability of Glc-rGO membrane, merely presenting the computation results. The authors claim that the nature of the interaction between Glc and rGO was identified; however, those kinds of interactions are certainly predictable without computation. For the experimental results, they still argue that there should be a proper concentration of Glc without convincing data; nevertheless, they said that the high Glc concentration induces a poor stability. In my opinion, the justification of such "proper concentration" is one of the major points of this manuscript, and the probable cause of the improved stability should be provided in experiments. At least, the interaction itself should be supported.

Response:

Thanks for pointing out this issue. We have provided additional experiments and discussions to address it.

For the interplay between GO and Glc, as suggested by the reviewer, we have performed experiments using UV-vis and XPS spectra to evidence the chemical affinity or certain interaction (hydrogen bonding) between GO and Glc (see **Supplementary Figure 11** in revised manuscript). We also performed experiments of Glc intercalation in GO laminates and GO membrane adsorption of Glc to demonstrate the affinity- or interaction-induced attraction between GO and Glc (see **Supplementary Figure 12** in revised manuscript). These results support the stabilization of laminates due to the affinity and interaction between GO and Glc. We also performed another experiment to evaluate the stability difference between GO and Glc-rGO membranes from macroscopic observation (see **Supplementary Figure 13** in revised manuscript).

For the proper concentration, we have modified and clarified the corresponding statements as the descriptions of the effects of intercalated Glc molecules on membrane performance. We provided discussions about the effects of Glc concentration on membrane performance in the sections based on the performance and Glc quantification data in **Figure 4** and **Figure 5**. From the experimental results of performance stability with varying pressure and quantification of Glc intercalation and leaching in the membranes, it implies that the Glc content used for reaction with GO had influences on the Glc intercalation and leaching of the laminates as well as the corresponding membrane performance.

With the additional experiments, we believe the interplay between GO and Glc can be supported. The modified statement of the effects of intercalated Glc is also supported by the experimental results of quantified Glc amount and membrane performance difference.

We have included the modifications and additional experimental results in revised manuscript and Supplementary Information:

Manuscript (RESULTS – Nanochannel confinement in water)

“Molecular dynamics (MD) simulations were performed to investigate the evolution of the laminate structure under water pressure conditions and to reveal the

contribution of Glc to the structural stability (**Figure 3a**, **Supplementary Figure 9**, and **Supplementary Movies 1–4**). rGO laminates with lower O/C ratios exhibited a more stable structure during a simulation of 50 ns, compared to GO, which is ascribed to the increased hydrophobicity of rGO. It was also found that the inserted Glc in the interlayers had a positive effect on improving the laminate stability. To identify the nature of the interaction between the Glc and rGO, a cluster model (**Supplementary Figure 10**) from the last frame of the MD simulation trajectory was employed by applying the independent gradient model (IGM) method. The 2D scatter plot and the isosurface ($\text{sign}(\lambda_2)\rho$) mapped intermolecular electron density gradient difference (δg_{inter}) for the model are shown in **Figure 3b**. The region close to $\text{sign}(\lambda_2)\rho = 0$ (green) denotes mainly the weak attractive van der Waals interactions. The cyan-to-blue region denotes relatively strong attractive interactions, including hydrogen bonds (**Figure 3c**). Thus, increasing the Glc amount in certain GO laminates stabilizes them to a certain extent. In addition, the interaction of GO and Glc was also experimentally evidenced (**Supplementary Figure 11**) which includes their chemical affinity and hydrogen bonding. As a result, Glc molecules could be favorably retained in laminates and adsorbed by laminates as observed in experiments (**Supplementary Figure 12**). We immersed the nylon substrate-supported GO and Glc-rGO laminates into water to preliminarily check their water stability (**Supplementary Figure 13**). The Glc-rGO laminates remained integrated while the GO laminates were seriously destructed by the water intrusion, confirming the improved macroscale stability of Glc-rGO. Therefore, through the chemical conversion process by Glc, GO laminates could be more sufficiently stable to have confined nanochannels for ion separation.”

“To further investigate the nanochannel deformation in laminates with the infiltrated pressurized flow (which is similar to pressure-driven filtration), the time-course pure water flux of different laminates at an applied pressure of 1 MPa was recorded (**Figure 3e**). With an externally driven force, water molecules can easily penetrate the nanochannels, resulting in a higher propensity for channel deformation. Rapid increases and subsequent decreases in the flux were observed for the GO, T0,

and T10 samples, correlating with the XRD analysis (**Figure 3d**). With longer reaction periods and higher Glc ratios, the flux decline was gradually minimized up to a reaction period of 60 min (T60) or Glc ratios of 10 (Glc10). For T80, Glc15, and Glc20, the decline again became obvious. This phenomenon is in accordance with our hypothesis that the included Glc molecules could be another cause of nanochannel deformation during pressurized filtration. Higher Glc ratio resulted in a more significant flux decline (samples Glc15 and Glc20 with higher Glc ratios used for laminate preparation in comparison to T80 with longer reaction periods) because the number of intercalated Glc molecules in the nanochannels would increase for Glc15 and Glc20. The optimal samples (Glc10 or T60) exhibited a minimal water flux decline, owing to the nanochannel constraint caused by the *optimal integration of GO frameworks and Glc intercalations*. Both the occupation of Glc molecules in the channels and the reduced interlayer spacing hindered water transport (simulation results in **Supplementary Figure 15** and **Supplementary Movies 5** and **6**), resulting in a decreased water flux, particularly when including a possibility of more Glc molecules in the nanochannel.”

Supplementary Information

“Supplementary Fig. 11. Characterizations of chemical affinity and interaction between GO and Glc. (a) Digital images showing suspensions and corresponding UV-vis spectra of GO (0.015 mg L⁻¹) and mixture of GO (0.015 mg L⁻¹) and Glc (0.15 mg L⁻¹); (b) XPS C1s spectra of GO and mixture of GO and Glc.”

“The suspension color of GO and Glc mixture (GO+Glc) became darker after a certain period even under room temperature (25 °C), with observed higher overall absorbance and peak shift due to the chemical affinity of GO and Glc; whereas that of GO was not significantly changed. In addition, there was a binding energy shift for C–O bond of GO after the Glc mixing from XPS C1s spectra, indicating the interaction between them and corresponding change of the chemical environment around the elements.”

“Supplementary Fig. 12. Glc intercalation in laminates and Glc adsorption by membranes. (a) Illustration of laminate fabrication with the filtration of suspensions of GO+Glc mixture and Glc10T60 and correspondingly quantified intercalation amount of Glc; (b) Illustration of Glc adsorption by GO and Glc10T60 membranes and correspondingly analyzed concentration change of Glc solution.”

“To measure the amount of Glc intercalated in the stacked laminates, the filtrate was collected after vacuum filtration of the suspension (chemically converted Glc10T60 and simple mixture of GO and Glc) with a known amount of added Glc for the preparation of the laminates. The filtrate was analyzed using the total organic carbon (TOC) technique to quantify the Glc. Hence, the amount of Glc intercalated in the laminates was obtained. Control samples of the pristine GO suspension (without Glc addition) were also prepared to exclude the TOC from the filtered impurities of materials other than Glc.

To measure the adsorption of Glc by GO-based membranes, the membrane was immersed in a 10 mL Glc solution with an initial concentration of 0.15 mg mL⁻¹. After 2 days, the Glc solution concentration was measured by TOC. Control samples of bare substrate were also prepared to exclude the adsorption by the substrate.

Glc could be intercalated in both GO laminates and chemically converted GO laminates. More Glc molecules could be retained in laminates after chemical conversion process than that of simple mixture. GO membranes and Glc10T60 membrane could adsorb certain Glc from the solution. These results support the favorable interaction or affinity between GO and Glc.”

“Supplementary Fig. 13. Macroscopic stability of laminate layer with water intrusion. Illustration of the water intrusion towards the laminates during immersion and surface morphology of GO and Glc-rGO (Glc10T60) after immersion.”

Manuscript (RESULTS – Nanochannel confinement in salt/water mixture)

“To further investigate laminar membranes with different nanochannel deformation constraint capabilities, four typical samples were selected for comparison: pristine GO, Glc10T20 (optimal Glc ratio but short reaction period), Glc10T60 (optimal), and Glc20T60 (optimal reaction period but excessive Glc ratio). The surface

zeta potentials of these membranes at neutral pH were all negative (**Figure 4c** and **Supplementary Figure 17**), which is desirable for the rejection of salt solutes containing high-valence anions (such as Na_2SO_4). The rejection of different salts by these membranes (**Figure 4d**) indicates that the ionic transport behavior is governed by the charge effect, leading to a high rejection of Na_2SO_4 and low rejection of MgCl_2 . However, the difference in rejection by *Glc10T60* and *Glc20T60* between the different salts narrowed significantly compared with that of *GO* and *Glc10T20*. The lower negative zeta potentials of *Glc10T60* and *Glc20T60* could explain their decreased attraction to high-valence cations, improving MgCl_2 rejection. Notably, stronger nanochannel confinement (an optimal example is *Glc10T60*) should improve the overall rejection of all types of salts by steric hindrance. However, the excessive use of *Glc* molecules (*Glc20T60*) resulted in unfavorable nanochannels for the separation and compromised salt rejection, indicating the adverse effects of the excessive amount of *Glc* used on the membrane performance. Our previously proposed ‘nanochannel-confined charged repulsion’¹⁹ for the *GO* nanochannel membrane emphasized the role of nanochannel confinement. Additionally, in this study, we determined the significance of the confinement constraint capability. Careful consideration should be given to the possible disruption of channeled nanosheets by intercalated substances.”

“Figure 4. (d) Rejection of salts by *GO*, *Glc10T20*, *Glc10T60*, and *Glc20T60* membranes. The error bars represent the standard deviations of three measurements for rejection performance and surface zeta potential.”

Manuscript (RESULTS – Role of the deformation constraint of nanochannel for salt/water separation)

“The improved hydrophobicity of these three Glc-rGO membranes (Supplementary Figure 14) indicates the improved water stability of their bulk structures. This was inconsistent with their nanochannel stability and salt rejection, as they had different responses to the infiltrated water flow in their different pore interiors. The amount of Glc retained after the preparation of these laminates could be different, considering the different degrees of GO reduction and interactions with Glc. The amount of Glc intercalated was then quantified and result was similar for the Glc10T20 and Glc10T60 laminates, whereas it was higher for Glc20T60 (Figure 5e, Supplementary Figure 12a and Supplementary Figure 20), indicating that higher amounts of Glc used in suspension lead to a higher possibility of retained Glc in the corresponding laminates. The higher Glc content in Glc20T60 laminates may be associated with its relatively poor performance. Although this explains the difference in performance between Glc10 and Glc20, the difference between T20 and T60 must be further investigated.

As Glc molecules have hydrophilic properties, the intercalated Glc molecules can also be dissolved with pressurized flow during the permeation test (Figure 5f). The leached amounts of Glc for these membranes during permeation tests were quantified in the order of Glc10T20 < Glc10T60 < Glc20T60 (5%, 15%, and 24%, respectively; Figure 5e). This order is also consistent with their flux decline tendency, indicating that the presence of Glc molecules in the nanochannels was related to channel stability. Higher Glc leaching indicates lower integration of Glc and rGO and, thus, a higher possibility of a perturbed structure during Glc leaching by pressurized flow. Therefore, the intercalation amount of Glc and the integration of Glc and GO for the laminates are influential for the nanochannel stability and membrane performance. The mutable and dynamic GO nanochannels cannot function like rigid pores in response to external pressure and NaCl in the feed, to strictly exclude hydrated ions from water. Nanochannel deformation frequently occurs when external conditions are changed. An improved deformation constraint can be obtained by optimizing the chemistry of the

GO nanosheets and avoiding the excessive intercalation of undesirable molecules into multilayered nanochannels to maximize the stable rejection of ions.”

“Figure 5. (e) Glc content in original membrane precursor suspension, intercalated Glc content in prepared membranes, and leached Glc content from membranes during permeation test. The error bars represent the standard deviations of three measurements.”

Comments from Referee 2

General Comment:

The authors have provided detailed responses that have done a commendable job of addressing my comments and comments from the other reviews. I was particularly pleased to see changes in some of the schematics, which I think will make the manuscript easier to interpret for readers. I do not have additional comments.

Response:

We thank the reviewer for taking the time and effort to review the manuscript.

Comments from Referee 4

General Comment:

The revised manuscript has been greatly improved. The authors have clarified all of my concerns. The present manuscript is suitable for publication in this journal.

Response:

We thank the reviewer for taking the time and effort to review the manuscript.

REVIEWERS' COMMENTS

Reviewer #2 (Remarks to the Author):

The author have revised the manuscript to satisfy previous reviewer comments. The authors have provided significant new experimental evidence of (1) the proper concentration of Glc required and (2) the nature of the interaction between Glc and rGO. In my opinion, the response regarding the "proper concentration" is sufficient. The explanation of the interaction between Glc and rGO remains vague, and I would suggest some minor edits as outlined below. The comments above are minor, and I should emphasize that the authors have done exceptional work running new experiments to improve the manuscript.

The authors should revise the new text in the "Nanochannel confinement in water" section. The added statement in Line 172 ("It was also found that the inserted Glc in the interlayers had a positive effect on improving the laminate stability") is unsupported. What was the measure of laminate stability in this case? The authors use molecular simulations to investigate the stability mechanism and conclude that van der Waals interactions are involved (this is obvious since van der Waals interactions are universal) and the authors only hint at hydrogen bonding being important. A stronger statement highlighting the main interactions should be given, if possible. The statement in Line 181: "In addition, the interaction of GO and Glc was also experimentally evidenced which includes their chemical affinity and hydrogen bonding" is unclear. The conclusion of hydrogen bonding based on the experiments in Supplementary Figure 11 is not explained, and I do not understand what is specifically meant by chemical affinity.

Responses to Reviewers' Comments

Manuscript ID: NCOMMS-22-31241B

Manuscript type: Article

Title: Deformation constraints of graphene oxide nanochannels under reverse osmosis

Corresponding author: Hideto Matsuyama; Wanqin Jin

We are grateful to the reviewers for their insightful comments on our paper, which are extremely helpful in improving the quality of the manuscript. We have been able to incorporate changes to reflect the suggestions provided by the reviewers. The changes within the manuscript are highlighted in blue.

Please find below a point-by-point response to the reviewers' comments and concerns.

Comments from Referee 2

- **General Comment:**

The author have revised the manuscript to satisfy previous reviewer comments. The authors have provided significant new experimental evidence of (1) the proper concentration of Glc required and (2) the nature of the interaction between Glc and rGO. In my opinion, the response regarding the "proper concentration" is sufficient. The explanation of the interaction between Glc and rGO remains vague, and I would suggest some minor edits as outlined below. The comments above are minor, and I should emphasize that the authors have done exceptional work running new experiments to improve the manuscript.

Response:

We thank the reviewer for the recognition of our previously revised manuscript and the provided valuable comments, which are very helpful in improving the quality of the research paper. We have revised manuscript according to the reviewer's comments. The changes within the manuscript are highlighted in blue. Please find below the response to the concern.

- **Comment 1:**

The authors should revise the new text in the "Nanochannel confinement in water" section. The added statement in Line 172 ("It was also found that the inserted Glc in the interlayers had a positive effect on improving the laminate stability") is unsupported. What was the measure of laminate stability in this case? The authors use molecular simulations to investigate the stability mechanism and conclude that van der Waals interactions are involved (this is obvious since van der Waals interactions are universal) and the authors only hint at hydrogen bonding being important. A stronger statement highlighting the main interactions should be given, if possible. The statement in Line 181: "In addition, the interaction of GO and Glc was also experimentally evidenced

which includes their chemical affinity and hydrogen bonding" is unclear. The conclusion of hydrogen bonding based on the experiments in Supplementary Figure 11 is not explained, and I do not understand what is specifically meant by chemical affinity.

Response:

Thanks for the comments. We have modified the context to clarify these concerns.

For the issue on **Line 172**, by mentioning “*It was also found that the inserted Glc in the interlayers had a positive effect on improving the laminate stability*”, we mean the better stability of Glc-rGO model found in computational simulations in Figure 3a. Therefore, we performed additional simulations and experimental characterizations to evidence the favorable interactions between Glc and GO which lead to the enhanced stability. We are sorry for the misunderstanding caused and we have modified the context to clarify this statement:

“Molecular dynamics (MD) simulations were performed to investigate the evolution of the laminate structure under water pressure conditions and to reveal the contribution of Glc to the structural stability (Figure 3a, Supplementary Figure 9, and Supplementary Movies 1–4). rGO laminates with lower O/C ratios exhibited a more stable structure during a simulation of 50 ns, compared to GO, which is ascribed to the increased hydrophobicity of rGO. This evidences the positive effect of chemical reduction. It was also found that the inserted Glc in the interlayers had a positive effect on improving the laminate stability, as the same rGO laminate model with more of intercalated Glc molecules exhibited better stability from the simulation results (Figure 3a, Glc(2)-rGO and Glc(6)-rGO). In this regard, the inserted Glc should have certain interaction with GO laminates. To identify the nature of the interaction...”

For the issue on the statement **highlighting the main interactions**, we have added a statement to claim the hydrogen bonding as the main interaction between inserted Glc and GO:

“...To identify the nature of the interaction between the Glc and rGO that resulted in the favorable intercalation, a cluster model (Supplementary Figure 11) from the last frame of the MD simulation trajectory was employed by applying the independent gradient model (IGM) method. The 2D scatter plot and the isosurface ($\text{sign}(\lambda_2)\rho$ mapped intermolecular electron density gradient difference (δg_{inter})) for the model are shown in Figure 3b. The region close to $\text{sign}(\lambda_2)\rho = 0$ (green) denotes mainly the weak attractive van der Waals interactions. The cyan-to-blue region denotes relatively strong attractive interactions, including hydrogen bonds (Figure 3c). It indicates that the hydrogen bonding was the main interaction between Glc and GO for their integration in the laminates. In addition...”

For the issue on **Line 181** and **Supplementary Figure 11 (Supplementary Figure 12** in revised manuscript), the chemical affinity we mentioned is referred to the chemical reducing property of Glc that can carry out chemical reaction with GO. We have modified the context as “chemical reduction role” and “hydrogen bonding role” to clarify this. We also provided brief explanations for the results in Supplementary Figure 11 (Supplementary Figure 12 in revised manuscript) that evidence the “chemical reduction role” and “hydrogen bonding role”, and detailed discussion was provided in the Supporting Information.

Below are the revisions made in this section and Supporting Information:

Manuscript (RESULTS – Nanochannel confinement in water)

“...To identify the nature of the interaction between the Glc and rGO that resulted in the favorable intercalation, a cluster model (Supplementary Figure 11) from the last frame of the MD simulation trajectory was employed by applying the independent gradient model (IGM) method. The 2D scatter plot and the isosurface ($\text{sign}(\lambda_2)\rho$ mapped intermolecular electron density gradient difference (δg_{inter})) for the model are shown in Figure 3b. The region close to $\text{sign}(\lambda_2)\rho = 0$ (green) denotes mainly the weak attractive van der Waals interactions. The cyan-to-blue region denotes relatively

*strong attractive interactions, including hydrogen bonds (Figure 3c). It indicates that the hydrogen bonding was the main interaction between Glc and GO for their integration in the laminates. In addition, the chemical reduction role and hydrogen bonding role of Glc were also experimentally evidenced by observed absorption peak shift in UV-vis spectra and chemical shift in XPS spectra of GO and Glc mixture (Supplementary Figure 12). As a result, Glc molecules could be favorably retained in laminates and adsorbed by laminates as observed in *corresponding* experiments (Supplementary Figure 13).*

Supporting Information (Supplementary Figure 12)

“The suspension color of GO and Glc mixture (GO+Glc) became darker after a certain period even under room temperature (25 °C), with observed higher overall absorbance and peak shift due to the chemical affinity of GO and Glc that leads to the chemical reduction of GO; whereas that of GO was not significantly changed. In addition, there was a binding energy shift for C–O bond of GO after the Glc mixing from XPS C1s spectra, indicating the interaction of hydrogen bonding between them with corresponding change of the chemical environment around the elements.”